# IDENTIFIABILITY RESULTS FOR MULTIMODAL CONTRASTIVE LEARNING

**Imant Daunhawer**[1,†], **Alice Bizeul**[1,2], **Emanuele Palumbo**[1,2], **Alexander Marx**[1,2,*] **& Julia E. Vogt**[1,*]

[1] Department of Computer Science, ETH Zurich
[2] ETH AI Center, ETH Zurich

## ABSTRACT

Contrastive learning is a cornerstone underlying recent progress in multi-view and multimodal learning, e.g., in representation learning with image/caption pairs. While its effectiveness is not yet fully understood, a line of recent work reveals that contrastive learning can invert the data generating process and recover ground truth latent factors shared between views. In this work, we present new identifiability results for multimodal contrastive learning, showing that it is possible to recover shared factors in a more general setup than the multi-view setting studied previously. Specifically, we distinguish between the multi-view setting with one generative mechanism (e.g., multiple cameras of the same type) and the multimodal setting that is characterized by *distinct* mechanisms (e.g., cameras and microphones). Our work generalizes previous identifiability results by redefining the generative process in terms of distinct mechanisms with modality-specific latent variables. We prove that contrastive learning can block-identify latent factors shared between modalities, even when there are nontrivial dependencies between factors. We empirically verify our identifiability results with numerical simulations and corroborate our findings on a complex multimodal dataset of image/text pairs. Zooming out, our work provides a theoretical basis for multimodal representation learning and explains in which settings multimodal contrastive learning can be effective in practice.

## 1 INTRODUCTION

Multimodal representation learning is an emerging field whose growth is fueled by recent developments in weakly-supervised learning algorithms and by the collection of suitable multimodal datasets. Multimodal data is characterized by the *co-occurence* of observations from two or more dependent data sources, such as paired images and captions (e.g., Salakhutdinov and Hinton, 2009; Shi et al., 2019; Radford et al., 2021), and more generally, multimodal observations are comprised of aligned measurements from different types of sensors (Baltrušaitis et al., 2019). Co-occurrence is a form of *weak supervision* (Shu et al., 2020; Locatello et al., 2020; Chen and Batmanghelich, 2020), in that paired observations can be viewed as proxies (i.e., weak labels) for a shared but unobserved ground truth factor. Among suitable representation learning methods for weakly supervised data, *contrastive learning* (Gutmann and Hyvärinen, 2010; Oord et al., 2018) stands out because it is designed to leverage co-occurring observations from different views. In practice, contrastive learning achieves promising results for multi-view and multimodal learning—a prominent example is the contribution of CLIP (Radford et al., 2021) to the groundbreaking advancements in text-to-image generation (Ramesh et al., 2021; 2022; Rombach et al., 2022; Saharia et al., 2022).

Despite its empirical success, it is not sufficiently well understood what explains the effectiveness of contrastive learning in practice. Recent works attribute its effectiveness to the recovery of shared latent factors from the underlying causal graph (Gresele et al., 2019; Zimmermann et al., 2021; von Kügelgen et al., 2021). From the perspective of multi-view independent component analysis, it was shown that contrastive learning can invert a nonlinear mixing function (i.e., a nonlinear generative process) that is applied to a latent variable with mutually independent components (Gresele et al., 2019; Zimmermann et al., 2021). More recently, von Kügelgen et al. (2021) show that contrastive learning can recover shared factors up to block-wise indeterminacies, even if there are nontrivial causal and statistical

---

*Joint authorship. †Correspondence to: `dimant@ethz.ch`.

dependencies between latent components. Collectively, these results suggest that contrastive learning can identify parts of an unknown data generating process from pairs of observations alone—even from high-dimensional multi-view observations with nontrivial dependencies. In our work, we investigate the identifiability of shared latent factors in the *multimodal* setting.

We consider a generative process with modality-specific mixing functions and modality-specific latent variables. Our design is motivated by the inherent heterogeneity of multimodal data, which follows naturally when observations are generated by different types of sensors (Baltrušaitis et al., 2019). For example, an agent can perceive its environment through distinct sensory modalities, such as cameras sensing light or microphones detecting sound waves. To model information that is shared between modalities, we take inspiration from the multi-view setting (von Kügelgen et al., 2021) and allow for nontrivial dependencies between latent variables. However, previous work only considers observations of the same data type and assumes that the same input leads to the same output across views. In this work, we introduce a model with *distinct generative mechanisms*, each of which can exhibit a significant degree of modality-specific variation. This distinction renders the multimodal setting more general compared to the multi-view setting considered by previous work.

In a nutshell, our work is concerned with *identifiability for multimodal representation learning* and focuses on *contrastive learning* as a particular algorithm for which we derive identifiability results. In Section 2, we cover relevant background on both topics, identifiability and contrastive learning. We then formalize the multimodal generative process as a latent variable model (Section 3) and prove that contrastive learning can block-identify latent factors shared between modalities (Section 4). We empirically verify the identifiability results with fully controlled numerical simulations (Section 5.1) and corroborate our findings on a complex multimodal dataset of image/text pairs (Section 5.2). Finally, we contextualize related literature (Section 6) and discuss potential limitations and opportunities for future work (Section 7).

## 2 PRELIMINARIES

### 2.1 IDENTIFIABILITY

Identifiability lies at the heart of many problems in the fields of independent component analysis (ICA), causal discovery, and inverse problems, among others (Lehmann and Casella, 2006). From the perspective of ICA, we consider the relation $\mathbf{x} = \mathbf{f}(\mathbf{z})$, where an observation $\mathbf{x}$ is generated from a mixing function $\mathbf{f}$ that is applied to a latent variable $\mathbf{z}$. The goal of ICA is to invert the mixing function in order to recover the latent variable *from observations alone*. In many settings, full identifiability is impossible and certain ambiguities might be acceptable. For example, identifiability might hold for a subset of components (i.e., partial identifiability). Typical ambiguities include permutation and element-wise transformations (i.e., component-wise indeterminacy), or identifiability up to groups of latent variables (i.e., block-wise indeterminacy). In the general case, when $\mathbf{f}$ is a nonlinear function, a landmark negative result states that the recovery of the latent variable given i.i.d. observations is fundamentally impossible (Hyvärinen and Pajunen, 1999). However, a recent line of pioneering works provides identifiability results for the difficult nonlinear case under additional assumptions, such as auxiliary variables (Hyvärinen and Morioka, 2017; Hyvärinen et al., 2019; Khemakhem et al., 2020) or multiple views (Gresele et al., 2019; Locatello et al., 2020; Zimmermann et al., 2021).

Most relevant to our investigation are previous works related to *multi-view nonlinear ICA* (Gresele et al., 2019; Lyu and Fu, 2020; Locatello et al., 2020; von Kügelgen et al., 2021; Lyu et al., 2022). Generally, this line of work considers the following generative process:

$$\mathbf{z} \sim p_{\mathbf{z}}, \quad \mathbf{x}_1 = \mathbf{f}_1(\mathbf{z}), \quad \mathbf{x}_2 = \mathbf{f}_2(\mathbf{z}), \tag{1}$$

where a latent variable, or a subset of its components, is shared between *pairs* of observations $(\mathbf{x}_1, \mathbf{x}_2) \sim p_{\mathbf{x}_1, \mathbf{x}_2}$, where the two views $\mathbf{x}_1$ and $\mathbf{x}_2$ are generated by two nonlinear mixing functions, $\mathbf{f}_1$ and $\mathbf{f}_2$ respectively. Intuitively, a second view can resolve ambiguity introduced by the nonlinear mixing, if both views contain a shared signal but are otherwise sufficiently distinct (Gresele et al., 2019). Previous works differ in their assumptions on the mixing functions and dependence relations between latent components. The majority of previous work considers mutually independent latent components (Song et al., 2014; Gresele et al., 2019; Locatello et al., 2020) or independent groups of shared and view-specific components (Lyu and Fu, 2020; Lyu et al., 2022). Moreover, some of these works (Song et al., 2014; Gresele et al., 2019; Lyu and Fu, 2020; Lyu et al., 2022) consider

view-specific[1] mixing functions. Venturing beyond the strict assumption of independent (groups of) components, von Kügelgen et al. (2021) consider additional causal and statistical dependencies between latents and show that the subset of shared components can be identified up to a block-wise indeterminacy. Our work considers heterogeneous modalities with causal and statistical dependencies between latents. We prove that shared factors can be block-identified in a novel setting with modality-specific mixing functions and modality-specific latent variables.

## 2.2 CONTRASTIVE LEARNING

Contrastive learning (Gutmann and Hyvärinen, 2010; Oord et al., 2018) is a self-supervised representation learning method that leverages weak supervision in the form of paired observations. On a high level, the method learns to distinguish "positive" pairs of encodings sampled from the joint distribution, against "negative" pairs sampled from the product of marginals. The popular InfoNCE objective (Oord et al., 2018) is defined as follows:

$$\mathcal{L}_{\text{InfoNCE}}(\mathbf{g}_1, \mathbf{g}_2) = \mathbb{E}_{\{\mathbf{x}_1^i, \mathbf{x}_2^i\}_{i=1}^K \sim p_{\mathbf{x}_1, \mathbf{x}_2}} \left[ -\sum_{i=1}^K \log \frac{\exp\{\text{sim}(\mathbf{g}_1(\mathbf{x}_1^i), \mathbf{g}_2(\mathbf{x}_2^i))/\tau\}}{\sum_{j=1}^K \exp\{\text{sim}(\mathbf{g}_1(\mathbf{x}_1^i), \mathbf{g}_2(\mathbf{x}_2^j))/\tau\}} \right], \quad (2)$$

where $\mathbf{g}_1$ and $\mathbf{g}_2$ are encoders for the first and second view, $\mathbf{x}_1$ and $\mathbf{x}_2$ respectively. It is common to use a single encoder $\mathbf{g}_1 = \mathbf{g}_2$ when $\mathbf{x}_2$ is an augmented version of $\mathbf{x}_1$ or when two augmentations are sampled from the same distribution to transform $\mathbf{x}_1$ and $\mathbf{x}_2$ respectively (e.g., Chen et al., 2020). The set of hyperparameters consists of the temperature $\tau$, a similarity metric $\text{sim}(\cdot, \cdot)$, and an integer $K$ that controls the number of negative pairs ($K - 1$) used for contrasting. The objective has an information-theoretic interpretation as a variational lower bound on the mutual information $I(\mathbf{g}_1(\mathbf{x}_1); \mathbf{g}_2(\mathbf{x}_2))$ (Oord et al., 2018; Poole et al., 2019) and it can also be interpreted as the alignment of positive pairs (numerator) with additional entropy regularization (denominator), where the regularizer disincentivizes a degenerate solution in which both encoders map to a constant (Wang and Isola, 2020). Formally, when instantiating the $\mathcal{L}_{\text{InfoNCE}}$ objective with $\tau = 1$ and $\text{sim}(a, b) = -(a - b)^2$, it asymptotically behaves like the objective

$$\mathcal{L}_{\text{AlignMaxEnt}}(\mathbf{g}) = \mathbb{E}_{(\mathbf{x}_1, \mathbf{x}_2) \sim p_{\mathbf{x}_1, \mathbf{x}_2}} \left[ \|\mathbf{g}(\mathbf{x}_1) - \mathbf{g}(\mathbf{x}_2)\|_2 \right] - H(\mathbf{g}(\mathbf{x})) \quad (3)$$

for a single encoder $\mathbf{g}$, when $K \to \infty$ (Wang and Isola, 2020; von Kügelgen et al., 2021).

In the setting with two heterogeneous modalities, it is natural to employ separate encoders $\mathbf{g}_1 \neq \mathbf{g}_2$, which can represent different architectures. Further, it is common to use a symmetrized version of the objective (e.g., see Zhang et al., 2022; Radford et al., 2021), which can be obtained by computing the mean of the loss in both directions:

$$\mathcal{L}_{\text{SymInfoNCE}}(\mathbf{g}_1, \mathbf{g}_2) = \tfrac{1}{2} \mathcal{L}_{\text{InfoNCE}}(\mathbf{g}_1, \mathbf{g}_2) + \tfrac{1}{2} \mathcal{L}_{\text{InfoNCE}}(\mathbf{g}_2, \mathbf{g}_1). \quad (4)$$

Akin to Equation (3), we can approximate the symmetrized objective for $\tau = 1$ and $\text{sim}(a, b) = -(a - b)^2$, with a large number of negative samples ($K \to \infty$), as follows:

$$\mathcal{L}_{\text{SymAlignMaxEnt}}(\mathbf{g}_1, \mathbf{g}_2) = \mathbb{E}_{(\mathbf{x}_1, \mathbf{x}_2) \sim p_{\mathbf{x}_1, \mathbf{x}_2}} \left[ \|\mathbf{g}_1(\mathbf{x}_1) - \mathbf{g}_2(\mathbf{x}_2)\|_2 \right] - \tfrac{1}{2} \left( H(\mathbf{g}_1(\mathbf{x}_1)) + H(\mathbf{g}_2(\mathbf{x}_2)) \right). \quad (5)$$

Since the similarity measure is symmetric, the approximation of the alignment term is identical for both $\mathcal{L}_{\text{InfoNCE}}(\mathbf{g}_1, \mathbf{g}_2)$ and $\mathcal{L}_{\text{InfoNCE}}(\mathbf{g}_2, \mathbf{g}_1)$. Each entropy term is approximated via the denominator of the respective loss term, which can be viewed as a nonparametric entropy estimator (Wang and Isola, 2020). For our experiments, we employ the finite-sample estimators $\mathcal{L}_{\text{InfoNCE}}$ and $\mathcal{L}_{\text{SymInfoNCE}}$, while for our theoretical analysis we use the estimand $\mathcal{L}_{\text{SymAlignMaxEnt}}$ to derive identifiability results.

## 3 GENERATIVE PROCESS

In the following, we formulate the multimodal generative process as a latent variable model (Section 3.1) and then specify our technical assumptions on the relation between modalities (Section 3.2).

---

[1]Note that we define *modality-specific* functions similar to the way Gresele et al. (2019), Lyu and Fu (2020), and Lyu et al. (2022) define *view-specific* functions. To clarify the distinction, we generally assume that observations from different modalities are generated by distinct mechanisms $\mathbf{f}_1 \neq \mathbf{f}_2$ with modality-specific latent variables, and we treat the multi-view setting as a special case, where $\mathbf{f}_1 = \mathbf{f}_2$ without view-specific latents.

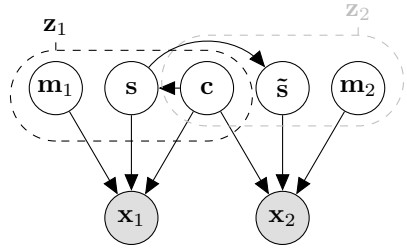

Figure 1: Illustration of the multimodal generative process. Latent variables are denoted by clear nodes and observations by shaded nodes. We partition the latent space into $\mathbf{z}_1 = (\mathbf{c}, \mathbf{s}, \mathbf{m}_1)$ and $\mathbf{z}_2 = (\tilde{\mathbf{c}}, \tilde{\mathbf{s}}, \mathbf{m}_2)$, where $\tilde{\mathbf{c}} = \mathbf{c}$ almost everywhere (Assumption 1) and hence we consider only $\mathbf{c}$. Further, $\tilde{\mathbf{s}}$ is a perturbed version of $\mathbf{s}$ (Assumption 2) and $\mathbf{m}_1, \mathbf{m}_2$ are modality-specific variables. The observations $\mathbf{x}_1$ and $\mathbf{x}_2$ are generated by two distinct mixing functions $\mathbf{f}_1 \neq \mathbf{f}_2$, which are applied to the subsets of latent variables $\mathbf{z}_1$ and $\mathbf{z}_2$ respectively.

### 3.1 LATENT VARIABLE MODEL

On a high level, we assume that there exists a continuous random variable $\mathbf{z}$ that takes values in the latent space $\mathcal{Z} \subseteq \mathbb{R}^n$, which contains all information to generate observations of both modalities.[2] Moreover, we assume that $\mathbf{z} = (\mathbf{c}, \mathbf{s}, \mathbf{m}_1, \mathbf{m}_2)$ can be uniquely partitioned into four disjoint parts:

*(i)* an invariant part $\mathbf{c}$ which is always shared across modalities, and which we refer to as *content*;

*(ii)* a variable part $\mathbf{s}$ which may change across modalities, and which we refer to as *style*;

*(iii)* two modality-specific parts, $\mathbf{m}_1$ and $\mathbf{m}_2$, each of which is unique to the respective modality.

Let $\mathbf{z}_1 = (\mathbf{c}, \mathbf{s}, \mathbf{m}_1)$ and $\mathbf{z}_2 = (\tilde{\mathbf{c}}, \tilde{\mathbf{s}}, \mathbf{m}_2)$, where $\tilde{\mathbf{c}} = \mathbf{c}$ almost everywhere and $\tilde{\mathbf{s}}$ is generated by perturbations that are specified in Section 3.2. Akin to multi-view ICA (Equation 1), we define the generative process for modalities $\mathbf{x}_1$ and $\mathbf{x}_2$ as follows:

$$\mathbf{z} \sim p_{\mathbf{z}}, \quad \mathbf{x}_1 = \mathbf{f}_1(\mathbf{z}_1), \quad \mathbf{x}_2 = \mathbf{f}_2(\mathbf{z}_2), \tag{6}$$

where $\mathbf{f}_1 : \mathcal{Z}_1 \to \mathcal{X}_1$ and $\mathbf{f}_2 : \mathcal{Z}_2 \to \mathcal{X}_2$ are two smooth and invertible mixing functions with smooth inverse (i.e., diffeomorphisms) that generate observations $\mathbf{x}_1$ and $\mathbf{x}_2$ taking values in $\mathcal{X}_1 \subseteq \mathbb{R}^{d_1}$ and $\mathcal{X}_2 \subseteq \mathbb{R}^{d_2}$ respectively. Generally, we assume that observations from different modalities are generated by distinct mechanisms $\mathbf{f}_1 \neq \mathbf{f}_2$ that take modality-specific latent variables as input. As for the multi-view setting (von Kügelgen et al., 2021), the considered generative process *goes beyond the classical ICA setting* by allowing for statistical dependencies within blocks of variables (e.g., between dimensions of $\mathbf{c}$) and also for causal dependencies from content to style, as illustrated in Figure 1. We assume that $p_{\mathbf{z}}$ is a smooth density that factorizes as $p_{\mathbf{z}} = p_{\mathbf{c}} \, p_{\mathbf{s}|\mathbf{c}} \, p_{\mathbf{m}_1} p_{\mathbf{m}_2}$ in the causal setting, and as the product of all involved marginals when there is no causal dependence from $\mathbf{c}$ to $\mathbf{s}$.

The outlined generative process is fairly general and it applies to a wide variety of practical settings. The content invariance describes a shared phenomenon that is not directly observed but manifests in the observations from both modalities. Style changes describe shared influences that are not robust across modalities, e.g., non-invertible transformations such as data augmentations, or non-deterministic effects of an unobserved confounder. Modality-specific factors can be viewed as variables that describe the inherent heterogeneity of each modality (e.g., background noise).

### 3.2 RELATION BETWEEN MODALITIES

Next, we specify our assumptions on the relation between modalities by defining the conditional distribution $p_{\mathbf{z}_2|\mathbf{z}_1}$, which describes the relation between latent variables $\mathbf{z}_1$ and $\mathbf{z}_2$, from which observations $\mathbf{x}_1$ and $\mathbf{x}_2$ are generated via Equation (6). Similar to previous work in the multi-view setting (von Kügelgen et al., 2021), we assume that content is invariant, i.e., $\tilde{\mathbf{c}} = \mathbf{c}$ almost everywhere (Assumption 1), and that $\tilde{\mathbf{s}}$ is a perturbed version of $\mathbf{s}$ (Assumption 2). To state our assumptions for the multimodal setting, we also need to consider the modality-specific latent variables.

---

[2] Put differently, we assume that all observations lie on a continuous manifold, which can have much smaller dimensionality than the observation space of the respective modality.

**Assumption 1** (Content-invariance). *The conditional density $p_{\mathbf{z}_2|\mathbf{z}_1}$ over $\mathcal{Z}_2 \times \mathcal{Z}_1$ takes the form*

$$p_{\mathbf{z}_2|\mathbf{z}_1}(\mathbf{z}_2|\mathbf{z}_1) = \delta(\tilde{\mathbf{c}} - \mathbf{c})p_{\tilde{\mathbf{s}}|\mathbf{s}}(\tilde{\mathbf{s}}|\mathbf{s})p_{\mathbf{m}_2}(\mathbf{m}_2) \tag{7}$$

*for some continuous density $p_{\tilde{\mathbf{s}}|\mathbf{s}}$ on $\mathcal{S} \times \mathcal{S}$, where $\delta(\cdot)$ is the Dirac delta function, i.e., $\tilde{\mathbf{c}} = \mathbf{c}$ a.e.*

To fully specify $p_{\mathbf{z}_2|\mathbf{z}_1}$, it remains to define the style changes, which are described by the conditional distribution $p_{\tilde{\mathbf{s}}|\mathbf{s}}$. There are several justifications for modeling such a stochastic relation between $\mathbf{s}$ and $\tilde{\mathbf{s}}$ (Zimmermann et al., 2021; von Kügelgen et al., 2021); one could either consider $\tilde{\mathbf{s}}$ to be a noisy version of $\mathbf{s}$, or consider $\tilde{\mathbf{s}}$ to be the result of an augmentation that induces a soft intervention on $\mathbf{s}$.[3]

**Assumption 2** (Style changes). *Let $\mathcal{A}$ be the powerset of style variables $\{1, \ldots, n_s\}$ and let $p_A$ be a distribution on $\mathcal{A}$. Then, the style conditional $p_{\tilde{\mathbf{s}}|\mathbf{s}}$ is obtained by conditioning on a set $A$:*

$$p_{\tilde{\mathbf{s}}|\mathbf{s}}(\tilde{\mathbf{s}}|\mathbf{s}) = \sum_{A \in \mathcal{A}} p_A(A) \left( \delta(\tilde{\mathbf{s}}_{A^c} - \mathbf{s}_{A^c})p_{\tilde{\mathbf{s}}_A|\mathbf{s}_A}(\tilde{\mathbf{s}}_A|\mathbf{s}_A) \right) \tag{8}$$

*where $p_{\tilde{\mathbf{s}}_A|\mathbf{s}_A}$ is a continuous density on $\mathcal{S}_A \times \mathcal{S}_A$, $\mathcal{S}_A \subseteq \mathcal{S}$ denotes the subspace of changing style variables specified by $A$, and $A^c = \{1, \ldots, n_s\} \backslash A$ denotes the complement of $A$.*

*Further, for any style variable $l \in \{1, \ldots, n_s\}$, there exists a set $A \subseteq \{1, \ldots, n_s\}$ with $l \in A$, s.t.*

*(i) $p_A(A) > 0$,*

*(ii) $p_{\tilde{\mathbf{s}}_A|\mathbf{s}_A}$ is smooth w.r.t. both $\mathbf{s}_A$ and $\tilde{\mathbf{s}}_A$, and*

*(iii) for any $\mathbf{s}_A$, $p_{\tilde{\mathbf{s}}_A|\mathbf{s}_A}(\cdot|\mathbf{s}_A) > 0$, in some open non-empty subset containing $\mathbf{s}_A$.*

Intuitively, to generate a pair of observations $(\mathbf{x}_1, \mathbf{x}_2)$, we independently flip a biased coin for each style dimension to select a subset of style features $A \subseteq \{1, \ldots, n_s\}$, which are jointly perturbed to obtain $\tilde{\mathbf{s}}$. Condition *(i)* ensures that every style dimension has a positive probability to be perturbed,[4] while *(ii)* and *(iii)* are technical smoothness conditions that will be used for the proof of Theorem 1.

Summarizing, in this section we have formalized the multimodal generative process as a latent variable model (Section 3.1) and specified our assumptions on the relation between modalities via the conditional distribution $p_{\mathbf{z}_1|\mathbf{z}_2}$ (Section 3.2). Next, we segue into the topic of representation learning and show that, for the specified generative process, multimodal contrastive learning can identify the content factors up to a block-wise indeterminacy.

## 4 IDENTIFIABILITY RESULTS

First, we need to define block-identifiability (von Kügelgen et al., 2021) for the multimodal setting in which we consider modality-specific mixing functions and encoders. In the following, $n_c$ denotes the number of content variables and the subscript $1{:}n_c$ denotes the subset of content dimensions (indexed from 1 to $n_c$ w.l.o.g.).

**Definition 1** (Block-identifiability). *The true content partition $\mathbf{c} = \mathbf{f}_1^{-1}(\mathbf{x}_1)_{1:n_c} = \mathbf{f}_2^{-1}(\mathbf{x}_2)_{1:n_c}$ is block-identified by a function $\mathbf{g}_i : \mathcal{X}_i \to \mathcal{Z}_i$, with $i \in \{1, 2\}$, if there exists an invertible function $\mathbf{h}_i : \mathbb{R}^{n_c} \to \mathbb{R}^{n_c}$, s.t. for the inferred content partition $\hat{\mathbf{c}}_i = \mathbf{g}_i(\mathbf{x}_i)_{1:n_c}$ it holds that $\hat{\mathbf{c}}_i = \mathbf{h}_i(\mathbf{c})$.*

It is important to note that block-identifiability does not require the identification of *individual* factors, which is the goal in multi-view nonlinear ICA (Gresele et al., 2019; Locatello et al., 2020; Zimmermann et al., 2021; Klindt et al., 2021) and the basis for strict definitions of disentanglement (Bengio et al., 2013; Higgins et al., 2018; Shu et al., 2020). Instead, our goal is to isolate the group of invariant factors (i.e., the content partition) from the remaining factors of variation in the data.

Specifically, our goal is to show that contrastive learning can block-identify the content variables for the multimodal setting described in Section 3. We formalize this in Theorem 1 and thereby relax the assumptions from previous work by allowing for distinct generating mechanisms $\mathbf{f}_1 \neq \mathbf{f}_2$ and additional modality-specific latent variables.

---

[3]Note that the asymmetry between $\mathbf{z}_1$ and $\mathbf{z}_2$ (or between $\mathbf{s}$ and $\tilde{\mathbf{s}}$) is not strictly required. We chose to write $\mathbf{z}_2$ as a perturbation of $\mathbf{z}_1$ to simplify the notation and for consistency with previous work. Instead, we could model *both* $\mathbf{z}_1$ and $\mathbf{z}_2$ via perturbations of $\mathbf{z}$, as described in Appendix A.2.

[4]If a style variable would be perturbed with zero probability, it would be a content variable.

| Generative process | | | $R^2$ (nonlinear) | | | Generative process | | | $R^2$ (nonlinear) | | |
|---|---|---|---|---|---|---|---|---|---|---|---|
| p(chg.) | Stat. | Cau. | Content c | Style s | | p(chg.) | Stat. | Cau. | Content c | Style s | Modality $m_i$ |
| 1.0 | ✗ | ✗ | **1.00** ± 0.00 | 0.00 ± 0.00 | | 1.0 | ✗ | ✗ | **0.99** ± 0.00 | 0.00 ± 0.00 | 0.00 ± 0.00 |
| 0.75 | ✗ | ✗ | **0.99** ± 0.01 | 0.00 ± 0.00 | | 0.75 | ✗ | ✗ | **1.00** ± 0.00 | 0.00 ± 0.00 | 0.00 ± 0.00 |
| 0.75 | ✓ | ✗ | **0.99** ± 0.00 | 0.52 ± 0.09 | | 0.75 | ✓ | ✗ | **0.95** ± 0.01 | 0.56 ± 0.23 | 0.00 ± 0.00 |
| 0.75 | ✗ | ✓ | **1.00** ± 0.00 | **0.79** ± 0.04 | | 0.75 | ✗ | ✓ | **0.98** ± 0.00 | **0.87** ± 0.04 | 0.00 ± 0.00 |
| 0.75 | ✓ | ✓ | **0.99** ± 0.01 | **0.81** ± 0.04 | | 0.75 | ✓ | ✓ | **0.95** ± 0.03 | **0.89** ± 0.07 | 0.00 ± 0.00 |

|                    |                     |
|--------------------|---------------------|
| (a) Original setting | (b) Multimodal setting |

Table 1: Results of the numerical simulations. We compare the original setting ($\mathbf{f}_1 = \mathbf{f}_2$, left table) with the multimodal setting ($\mathbf{f}_1 \neq \mathbf{f}_2$, right table). Only the multimodal setting includes modality-specific latent variables. Each row presents the results of a different setup with varying style-change probability p(chg.) and possible statistical (Stat.) and/or causal (Caus.) dependencies. Each value denotes the $R^2$ coefficient of determination (averaged across 3 seeds) for a nonlinear regression model that predicts the respective ground truth factor ($\mathbf{c}, \mathbf{s},$ or $\mathbf{m}_i$) from the learned representation.

**Theorem 1.** *Assume the data generating process described in Sec. 3.1, i.e. data pairs $(\mathbf{x}_1, \mathbf{x}_2)$ generated from Equation (6) with $p_{\mathbf{z}_1} = p_{\mathbf{z} \setminus \{\mathbf{m}_2\}}$ and $p_{\mathbf{z}_2 | \mathbf{z}_1}$ as defined in Assumptions 1 and 2. Further, assume that $p_{\mathbf{z}}$ is a smooth and continuous density on $\mathcal{Z}$ with $p_{\mathbf{z}}(\mathbf{z}) > 0$ almost everywhere. Let $\mathbf{g}_1 : \mathcal{X}_1 \to (0,1)^{n_c}$ and $\mathbf{g}_2 : \mathcal{X}_2 \to (0,1)^{n_c}$ be smooth functions that minimize $\mathcal{L}_{SymAlignMaxEnt}$ as defined in Eq. (5). Then, $\mathbf{g}_1$ and $\mathbf{g}_2$ block-identify the true content variables in the sense of Def. 1.*

A proof of Theorem 1 is provided in Appendix A.1. Intuitively, the result states that contrastive learning can identify the content variables up to a block-wise indeterminacy. Similar to previous work, the result is based on the optimization of the asymptotic form of the contrastive loss (Equation 5). Moreover, Theorem 1 assumes that the number of content variables is known or that it can be estimated (e.g., with a heuristic like the elbow method). We address the question of selecting the encoding size with dimensionality ablations throughout our experiments. In Section 7, we will return to the discussion of the assumptions in the context of the experimental results.

## 5 EXPERIMENTS

The goal of our experiments is to test whether contrastive learning can block-identify content in the multimodal setting, as described by Theorem 1. First, we verify identifiability in a fully controlled setting with numerical simulations (Section 5.1). Second, we corroborate our findings on a complex multimodal dataset of image/text pairs (Section 5.2). The code is provided in our github repository.[5]

### 5.1 NUMERICAL SIMULATION

We extend the numerical simulation from von Kügelgen et al. (2021) and implement the multimodal setting using modality-specific mixing functions ($\mathbf{f}_1 \neq \mathbf{f}_2$) with modality-specific latent variables. The numerical simulation allows us to measure identifiability with full control over the generative process. The data generation is consistent with the generative process described in Section 3. We sample $\mathbf{c} \sim \mathcal{N}(0, \Sigma_{\mathbf{c}})$, $\mathbf{m}_i \sim \mathcal{N}(0, \Sigma_{\mathbf{m}_i})$, and $\mathbf{s} \sim \mathcal{N}(\mathbf{a} + B\mathbf{c}, \Sigma_{\mathbf{s}})$. Statistical dependencies within blocks (e.g., among components of $\mathbf{c}$) are induced by non-zero off-diagonal entries in the corresponding covariance matrix (e.g. in $\Sigma_{\mathbf{c}}$). To induce a causal dependence from content to style, we set $a_i, B_{ij} \sim \mathcal{N}(0, 1)$; otherwise, we set $a_i, B_{ij} = 0$. For style changes, Gaussian noise is added with probability $\pi$ independently for each style dimension: $\tilde{\mathbf{s}}_i = \mathbf{s}_i + \epsilon$, where $\epsilon \sim \mathcal{N}(0, \Sigma_\epsilon)$ with probability $\pi$. We generate the observations $\mathbf{x}_1 = \mathbf{f}_1(\mathbf{c}, \mathbf{s}, \mathbf{m}_1)$ and $\mathbf{x}_2 = \mathbf{f}_2(\mathbf{c}, \tilde{\mathbf{s}}, \mathbf{m}_2)$ using two *distinct* nonlinear mixing functions, i.e, for each $i \in \{1, 2\}$, $\mathbf{f}_i : \mathbb{R}^d \to \mathbb{R}^d$ is a separate, invertible 3-layer MLP with LeakyReLU activations. We train the encoders for 300,000 iterations using the symmetrized InfoNCE objective (Equation 4) and the hyperparameters listed in Appendix B.1. We evaluate block-identifiability by predicting the ground truth factors from the learned representation using kernel ridge regression and report the $R^2$ coefficient of determination on holdout data.

---

[5]https://github.com/imantdaunhawer/multimodal-contrastive-learning.

**Results** We compare the original setting ($\mathbf{f}_1 = \mathbf{f}_2$, Table 1a) with the multimodal setting ($\mathbf{f}_1 \neq \mathbf{f}_2$, Table 1b) and find that content can be block-identified in *both* settings, as the $R^2$ score is close to one for the prediction of content, and quasi-random for the prediction of style and modality-specific information. Consistent with previous work, we observe that some style information can be predicted when there are statistical and/or causal dependencies; this is expected because statistical dependencies decrease the effective dimensionality of content, while the causal dependence $\mathbf{c} \rightarrow \mathbf{s}$ makes style partially predictable from the encoded content information. Overall, the results of the numerical simulation are consistent with our theoretical result from Theorem 1, showing that contrastive learning can block-identify content in the multimodal setting.

## 5.2 Image/text pairs

Next, we test whether block-identifiability holds in a more realistic setting with image/text pairs—two complex modalities with distinct generating mechanisms. We extend the *Causal3DIdent* dataset (von Kügelgen et al., 2021; Zimmermann et al., 2021), which allows us to measure and control the ground truth latent factors used to generate complex observations. We use *Blender* (Blender Online Community, 2018) to render high-dimensional images that depict a scene with a colored object illuminated by a differently colored spotlight and positioned in front of a colored background. The scene is defined by 11 latent factors: the shape of the object (7 classes), position of the object ($x, y, z$ coordinates), orientation of the object ($\alpha, \beta, \gamma$ angles), position of the spotlight ($\theta$ angle), as well as the color of the object, background, and spotlight respectively (one numerical value for each).

*Multimodal3DIdent* We extend the *Causal3DIdent* dataset to the multimodal setting as follows. We generate textual descriptions from the latent factors by adapting the text rendering from the *CLEVR* dataset (Johnson et al., 2017). Each image/text pair shares information about the shape of the object (cow, teapot, etc.) and its position in the scene (e.g., bottom-right). For each position factor, we use three clearly discernable values (top/center/bottom; left/center/right), which can be described in text more naturally than coordinates. While shape and position are always shared (i.e., content) be-

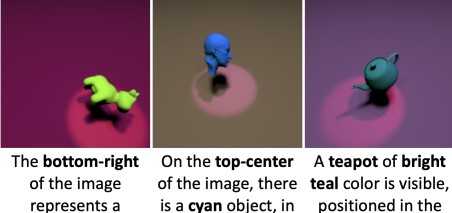

The **bottom-right** of the image represents a **lawngreen** colored **cow**.

On the **top-center** of the image, there is a **cyan** object, in the form of a **head**.

A **teapot** of **bright teal** color is visible, positioned in the **mid-center** of the image.

Figure 2: Examples of image/text pairs.

tween the paired image and text, the color of the object is causally influenced by position and is stochastically shared (i.e., style). For the object color, we use a continuous hue value, whereas for the text we match the RGB value with the nearest value from a given palette (i.e., a list of named colors, such as brown, beige, olive, etc.). The color palette is randomly sampled from a set of three palettes to ensure the object color depicted in the image does not uniquely determine the color described in the text. As modality-specific factors for the images, we have object rotation, spotlight position, and background color, while for the textual descriptions, we follow Johnson et al. (2017) and use 5 different types of phrases to introduce modality-specific variation. Examples of image/text pairs are shown in Figure 2. Further details about the dataset are provided in Appendix B.1.

We train the encoders for 100,000 iterations using the symmetrized InfoNCE objective (Equation 4) and the hyperparameters listed in Appendix B.1. For the image encoder we use a ResNet-18 architecture (He et al., 2016) and for the text we use a convolutional network. As for the numerical simulation, we evaluate block-identifiability by predicting the ground truth factors from the learned representation. For continuous factors, we use kernel ridge regression and report the $R^2$ score, whereas for discrete factors we report the classification accuracy of an MLP with a single hidden layer.

**Results** Figure 3 presents the results on *Multimodal3DIdent* with a dimensionality ablation, where we vary the size of the encoding of the model. Content factors (object position and shape) are always encoded well, unless the encoding size is too small (i.e., smaller than 3-4 dimensions). When there is sufficient capacity, style information (object color) is also encoded, partly because there is a causal dependence from content to style and partly because of the excess capacity, as already observed in previous work. Image-specific information (object rotation, spotlight position, background color) is mostly discarded, independent of the encoding size. Text-specific information (phrasing) is encoded to a moderate degree (48–80% accuracy), which we attribute to the fact that phrasing is a discrete factor that violates the assumption of continuous latents. This hints at possible limitations in the

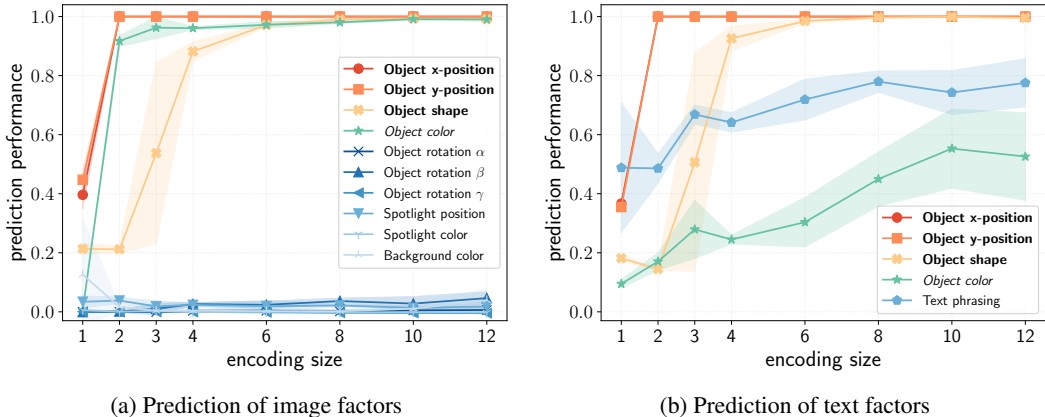

(a) Prediction of image factors         (b) Prediction of text factors

Figure 3: Results on *Multimodal3DIdent* as a function of the encoding size of the model. We assess the nonlinear prediction of ground truth image factors (left subplot) and text factors (right subplot) to quantify how well the learned representation encodes the respective factors. Content factors are denoted in bold and style factors in italic. Along the x-axis, we vary the encoding size, i.e., the output dimensionality of the model. We measure the prediction performance in terms of the $R^2$ coefficient of determination for continuous factors and classification accuracy for discrete factors respectively. Each point denotes the average across three seeds and bands show one standard deviation.

presence of discrete latent factors, which we further investigate in Appendix B.2 and discuss in Section 7. Overall, our results suggest that contrastive learning can block-identify content factors in a complex multimodal setting with image/text pairs.

## 6 RELATED WORK

**Multi-view nonlinear ICA** The goal of multi-view nonlinear ICA is to identify latent factors shared between different views, as described in Section 2.1. There is a thread of works (Gresele et al., 2019; Locatello et al., 2020) that recover the latent variable up to a component-wise indeterminacy in a setting with mutually independent latent components, or up to block-wise inderterminacies in the case of independent groups of shared and view-specific components (Lyu and Fu, 2020; Lyu et al., 2022). Beyond the assumption of independent (groups of) components, there is a line of works (von Kügelgen et al., 2021; Kong et al., 2022) that partition the latent space into blocks of invariant and blocks of changing components and show that the invariant components can be identified up to a block-wise indeterminacy, even when there are nontrivial dependencies between latent components. Our work advances in this direction and considers heterogeneous modalities with nontrivial statistical and causal dependencies between latents. We prove that shared factors can be block-identified in a novel setting with modality-specific mixing functions and modality-specific latent variables.

**Multimodal representation learning** Multimodal representation learning seeks to integrate information from heterogeneous sources into a joint representation (Baltrušaitis et al., 2019; Guo et al., 2019). There is a myriad of methods designed to learn representations of multimodal data either directly or indirectly. Among methods that learn representations indirectly, there are multimodal autoencoders (Ngiam et al., 2011; Geng et al., 2022; Bachmann et al., 2022; Aghajanyan et al., 2022) and a large variety of multimodal generative models (e.g., Suzuki et al., 2016; Wu and Goodman, 2018; Shi et al., 2019; Huang et al., 2018; Tsai et al., 2019; Ramesh et al., 2021) that learn representations by backpropagation of different forms of reconstruction and/or masked prediction error. A more direct approach is taken by decoder-free methods that maximize the similarity between the encodings of different modalities. This class of methods includes nonlinear canonical correlation analysis (Akaho, 2001; Bach and Jordan, 2002; Andrew et al., 2013; Wang et al., 2016; Tang et al., 2017; Karami and Schuurmans, 2021) as well as multi-view and multimodal contrastive learning (Tian et al., 2019; Bachman et al., 2019; Federici et al., 2020; Tsai et al., 2021; Radford et al., 2021; Poklukar et al., 2022). While all of the named methods aim to integrate information across modalities, they do not answer the underlying question of identifiability, which our work seeks to address.

## 7 DISCUSSION

**Implications and scope**  We have shown that contrastive learning can block-identify shared factors in the multimodal setting. Numerical simulations (Section 5.1) verify our main theoretical result (Theorem 1), showing that contrastive learning block-identifies content information (Definition 1), when the size of the encoding matches the number of content factors. Experiments on a complex dataset of image/text pairs corroborate that contrastive learning can isolate content in a more realistic setting and even under some violations of the assumptions underlying Theorem 1. In Appendix B.2, we include further experiments that test violations with discrete factors and dimensionality ablations that examine the robustness and sample complexity. More generally, we observe that contrastive learning encodes invariant information (i.e., content) very well across all settings. When there is sufficient capacity, stochastically shared information (i.e., style) is encoded to a moderate degree, but without affecting the prediction of invariant information. For practice, our results suggest that contrastive learning without capacity constraints can encode *any* shared factor, irregardless of whether the factor is truly invariant across modalities or if its effect on the observations is confounded by noise or other factors. This is in line with the information-theoretic view (Oord et al., 2018; Poole et al., 2019), i.e., that contrastive learning maximizes the mutual information between representations—a measure of mutual dependence that quantifies *any* information that is shared. Our results demonstrate that the size of the encoding can be reduced to learn a representation that recovers invariant information, as captured by the notion of block-identifiability. In practice, this can be leveraged for representation learning in settings of content-preserving distribution shifts (Mitrovic et al., 2021; Federici et al., 2021), where information relevant for a downstream task remains unchanged.

**Limitations and outlook**  First, Theorem 1 suggests that only *invariant* factors can be block-identified. However, in practice, there can be pairs of observations for which the invariance is inadvertently violated, e.g., due to measurement errors, occlusions, or other mistakes in the data collection. On the one hand, such a violation can be viewed as a mere artifact of the data collection and could be managed via interventions on the generative process, e.g., actions in reinforcement learning (Lippe et al., 2022; Brehmer et al., 2022; Ahuja et al., 2022; Lachapelle et al., 2022). On the other hand, violations of the content-invariance blur the line between content and style factors and it would be interesting to study identifiability in a more general setting with *only* stochastically shared factors. Second, Theorem 1 assumes that the number of content factors is known or that it can be estimated. In practice, this might not be a significant limitation, since the number of content factors can be viewed as a single hyperparameter (e.g., Locatello et al., 2020), though the design of suitable heuristics is an interesting research direction. We explore the idea of estimating the number of content factors in Appendix B.2 Figure 7. Third, Theorem 1 assumes that all latent factors are continuous. While this assumption prevails in related work (Hyvärinen and Pajunen, 1999; Hyvärinen and Morioka, 2016; Hyvärinen et al., 2019; Gresele et al., 2019; Locatello et al., 2019; 2020; Zimmermann et al., 2021; von Kügelgen et al., 2021; Klindt et al., 2021), our results in Figure 3b indicate that in the presence of discrete factors, some style or modality-specific information can be encoded. In Appendix B.2 Figure 5, we provide numerical simulations that support these findings. Finally, our model can be extended to more than two modalities—a setting for which there are intriguing identifiability results (Gresele et al., 2019; Schölkopf et al., 2016) as well as suitable learning objectives (Tian et al., 2019; Lyu et al., 2022). Summarizing, the described limitations mirror the assumptions on the generative process (Section 3), which may be relaxed in future work.

## 8 CONCLUSION

We addressed the problem of identifiability for multimodal representation learning and showed that contrastive learning can block-identify latent factors shared between heterogeneous modalities. We formalize the multimodal generative process as a novel latent variable model with modality-specific generative mechanisms and nontrivial statistical and causal dependencies between latents. We prove that contrastive learning can identify shared latent factors up to a block-wise indeterminacy and therefore isolate invariances between modalities from other changeable factors. Our theoretical results are corroborated by numerical simulations and on a complex multimodal dataset of image/text pairs. More generally, we believe that our work will help in shaping a theoretical foundation for multimodal representation learning and that further relaxations of the presented generative process offer rich opportunities for future work.

## ACKNOWLEDGEMENTS

ID was supported by the SNSF grant *#200021-188466*. EP was supported by the grant *#2021-911* of the Strategic Focal Area "Personalized Health and Related Technologies (PHRT)" of the ETH Domain (Swiss Federal Institutes of Technology). Experiments were performed on the ETH Zurich Leonhard cluster. Special thanks to Kieran Chin-Cheong for his support in the early stages of the project as well as to Luigi Gresele and Julius von Kügelgen for helpful discussions.

## REPRODUCIBILITY STATEMENT

For our theoretical statements, we provide detailed derivations and state the necessary assumptions. The generative process is specified in Section 3 and the assumptions for block-identifiability are referenced in Theorem 1. We test violations of the key assumptions with suitable experiments (dimensionality ablations; discrete latent factors) and discuss the limitations of our work in Section 7. Further, we empirically verify our theoretical results with numerical simulations and on complex multimodal data. To ensure empirical reproducibility, the results of every experiment were averaged over multiple seeds and are reported with standard deviations. Information about implementation details, hyperparameter settings, and evaluation metrics are included in Appendix B.1. Additionally, we publish the code to reproduce the experiments.

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

# A  THEORY

## A.1  PROOF OF THEOREM 1

**Theorem 1.** *Assume the data generating process described in Sec. 3.1, i.e. data pairs $(\mathbf{x}_1, \mathbf{x}_2)$ generated from Equation (6) with $p_{\mathbf{z}_1} = p_{\mathbf{z}\setminus\{\mathbf{m}_2\}}$ and $p_{\mathbf{z}_2|\mathbf{z}_1}$ as defined in Assumptions 1 and 2. Further, assume that $p_{\mathbf{z}}$ is a smooth and continuous density on $\mathcal{Z}$ with $p_{\mathbf{z}}(\mathbf{z}) > 0$ almost everywhere. Let $\mathbf{g}_1 : \mathcal{X}_1 \to (0,1)^{n_c}$ and $\mathbf{g}_2 : \mathcal{X}_2 \to (0,1)^{n_c}$ be smooth functions that minimize $\mathcal{L}_{SymAlignMaxEnt}$ as defined in Eq. (5). Then, $\mathbf{g}_1$ and $\mathbf{g}_2$ block-identify the true content variables in the sense of Def. 1.*

*Proof.* To prove Theorem 1, we follow the proof structure from von Kügelgen et al. (2021, Theorem 4.4) and divide the proof into three steps. First, we show that there exists a pair of smooth functions $\mathbf{g}_1^*, \mathbf{g}_2^*$ that attain the global minimum of $\mathcal{L}_{SymAlignMaxEnt}$ (Eq. 5). Further, in Equations (13–15), we derive invariance conditions that have to hold almost surely for any pair of smooth functions $\mathbf{g}_1, \mathbf{g}_2$ attaining the global minimum of Eq. (5). In Step 2, we use the invariance conditions derived in Step 1 to show by contradiction that *any* pair of smooth functions $\mathbf{g}_1, \mathbf{g}_2$ that attain the global minimum in Eq. (5) can only depend on content and not on style or modality-specific information. In the third and final step, for $\mathbf{h}_1 := \mathbf{g}_1 \circ \mathbf{f}_1$ and $\mathbf{h}_2 := \mathbf{g}_2 \circ \mathbf{f}_2$, we show that both functions must be bijections and hence that $\mathbf{c}$ is block-identified by $\mathbf{g}_1$ and $\mathbf{g}_2$ respectively.

**Step 1.**  Recall the asymptotic form of the objective, as defined in Equation (5):

$$\mathcal{L}_{SymAlignMaxEnt}(\mathbf{g}_1, \mathbf{g}_2) = \mathbb{E}_{(\mathbf{x}_1,\mathbf{x}_2)\sim p_{\mathbf{x}_1,\mathbf{x}_2}}\left[\|\mathbf{g}_1(\mathbf{x}_1) - \mathbf{g}_2(\mathbf{x}_2)\|_2\right] - \tfrac{1}{2}\left(H(\mathbf{g}_1(\mathbf{x}_1)) + H(\mathbf{g}_2(\mathbf{x}_2))\right) . \quad (5)$$

The global minimum of $\mathcal{L}_{SymAlignMaxEnt}$ is reached when the first term is minimized and the second term is maximized. The first term is minimized when the encoders $\mathbf{g}_1$ and $\mathbf{g}_2$ are perfectly aligned, i.e., when $\mathbf{g}_1(\mathbf{x}_1) = \mathbf{g}_2(\mathbf{x}_2)$ holds for all pairs $(\mathbf{x}_1, \mathbf{x}_2) \sim p_{\mathbf{x}_1,\mathbf{x}_2}$. The second term attains its maximum when $\mathbf{g}_1$ and $\mathbf{g}_2$ map to a uniformly distributed random variable on $(0,1)^{n_c}$ respectively.[6]

To show that there *exists* a pair of functions that minimize $\mathcal{L}_{SymAlignMaxEnt}$, let $\mathbf{g}_1^* := \boldsymbol{d}_1 \circ \mathbf{f}_{1,1:n_c}^{-1}$ and let $\mathbf{g}_2^* := \boldsymbol{d}_2 \circ \mathbf{f}_{2,1:n_c}^{-1}$, where the subscript $1{:}n_c$ indexes the subset of content dimensions w.l.o.g. and where $\boldsymbol{d}_1$ and $\boldsymbol{d}_2$ will be defined using the Darmois construction (Darmois, 1951; Hyvärinen and Pajunen, 1999). First, recall that $\mathbf{f}_1^{-1}(\mathbf{x}_1)_{1:n_c} = \mathbf{c}$ and that $\mathbf{f}_2^{-1}(\mathbf{x}_2)_{1:n_c} = \tilde{\mathbf{c}}$ by definition. Second, for $i \in \{1, 2\}$, let us define $\boldsymbol{d}_i : \mathcal{C} \mapsto (0,1)^{n_c}$ using the Darmois construction, such that $\boldsymbol{d}_i$ maps $\mathbf{c}$ and $\tilde{\mathbf{c}}$ to a uniform random variable respectively. It follows that $\mathbf{g}_1^*, \mathbf{g}_2^*$ are smooth functions, because any function $\boldsymbol{d}_i$ obtained via the Darmois construction is smooth and $\mathbf{f}_1^{-1}, \mathbf{f}_2^{-1}$ are smooth as well (each being the inverse of a smooth function).

Next, we show that the pair of functions $\mathbf{g}_1^*, \mathbf{g}_2^*$, as defined above, attains the global minimum of the objective $\mathcal{L}_{SymAlignMaxEnt}$. We have that

$$\mathcal{L}_{SymAlignMaxEnt}(\mathbf{g}_1^*, \mathbf{g}_2^*) = \mathbb{E}_{(\mathbf{x}_1,\mathbf{x}_2)\sim p_{\mathbf{x}_1,\mathbf{x}_2}}\left[\|\mathbf{g}_1^*(\mathbf{x}_1) - \mathbf{g}_2^*(\mathbf{x}_2)\|_2\right] - \tfrac{1}{2}\left(H(\mathbf{g}_1^*(\mathbf{x}_1)) + H(\mathbf{g}_2^*(\mathbf{x}_2))\right) \quad (9)$$

$$= \mathbb{E}_{(\mathbf{x}_1,\mathbf{x}_2)\sim p_{\mathbf{x}_1,\mathbf{x}_2}}\left[\|\boldsymbol{d}_1(\mathbf{c}) - \boldsymbol{d}_2(\tilde{\mathbf{c}})\|_2\right] - \tfrac{1}{2}\left(H(\boldsymbol{d}_1(\mathbf{c})) + H(\boldsymbol{d}_2(\tilde{\mathbf{c}}))\right) \quad (10)$$

$$= 0 \quad (11)$$

where by Assumption 1, $\mathbf{c} = \tilde{\mathbf{c}}$ almost surely, which implies that the first term is zero almost surely. Further, $\boldsymbol{d}_i$ maps $\mathbf{c}, \tilde{\mathbf{c}}$ to uniformly distributed random variables on $(0,1)^{n_c}$, which implies that the differential entropy of $\boldsymbol{d}_1(\mathbf{c})$ and $\boldsymbol{d}_2(\tilde{\mathbf{c}})$ is zero, as well. Consequently, there exists a pair of functions $\mathbf{g}_1^*, \mathbf{g}_2^*$ that minimizes $\mathcal{L}_{SymAlignMaxEnt}$.

Next, let $\mathbf{g}_1 : \mathcal{X}_1 \mapsto (0,1)^{n_c}$ and $\mathbf{g}_2 : \mathcal{X}_2 \mapsto (0,1)^{n_c}$ be *any* pair of smooth functions that attains the global minimum of Eq. (5), i.e.,

$$\mathcal{L}_{SymAlignMaxEnt}(\mathbf{g}_1, \mathbf{g}_2) = \mathbb{E}_{(\mathbf{x}_1,\mathbf{x}_2)\sim p_{\mathbf{x}_1,\mathbf{x}_2}}\left[\|\mathbf{g}_1(\mathbf{x}_1) - \mathbf{g}_2(\mathbf{x}_2)\|_2\right] - \tfrac{1}{2}\left(H(\mathbf{g}_1(\mathbf{x}_1)) + H(\mathbf{g}_2(\mathbf{x}_2))\right) = 0 .$$
$$(12)$$

Let $\mathbf{h}_1 := \mathbf{g}_1 \circ \mathbf{f}_1$ and $\mathbf{h}_2 := \mathbf{g}_2 \circ \mathbf{f}_2$, and notice that both are smooth functions since all involved functions are smooth by definition. Since Equation (12) is a global minimum, it implies the following

---

[6]Note that we restrict the range of $\mathbf{g}_1$ and $\mathbf{g}_2$ to $(0,1)^{n_c}$ by definition merely to simplify the notation. Generally, the uniform distribution $\mathcal{U}(a, b)$ is the maximum entropy distribution on the interval $[a, b]$.

invariance conditions for the individual terms:

$$\mathbb{E}_{(\mathbf{x}_1, \mathbf{x}_2) \sim p_{\mathbf{x}_1, \mathbf{x}_2}} \left[\|\mathbf{h}_1(\mathbf{z}_1) - \mathbf{h}_2(\mathbf{z}_2)\|_2\right] = 0 \tag{13}$$

$$H(\mathbf{h}_1(\mathbf{z}_2)) = 0 \tag{14}$$

$$H(\mathbf{h}_2(\mathbf{z}_2)) = 0 \tag{15}$$

Hence, $\mathbf{h}_1(\mathbf{z}_1) = \mathbf{h}_2(\mathbf{z}_2)$ must hold almost surely w.r.t. $p_{\mathbf{x}_1, \mathbf{x}_2}$. Additionally, Equation (14) (resp. Equation (15)) implies that $\hat{\mathbf{c}}_1 = \mathbf{h}_1(\mathbf{z}_1)$ (resp. $\hat{\mathbf{c}}_2 = \mathbf{h}_2(\mathbf{z}_2)$) must be uniform on $(0, 1)^{n_c}$.

**Step 2.** Next, we show that any pair of functions that minimize $\mathcal{L}_{\text{SymAlignMaxEnt}}$ depend only on content information. Since style is independent of $\mathbf{m}_1$ and $\mathbf{m}_2$, we first show that $\mathbf{h}_1(\mathbf{z}_1)$ does not depend on $\mathbf{m}_1$, and that $\mathbf{h}_2(\mathbf{z}_2)$ does not depend on $\mathbf{m}_2$. We then show that $\mathbf{h}_1$ and $\mathbf{h}_2$ also cannot depend on style, based on a result from previous work.

First note, that we can exclude all degenerate solutions where $\mathbf{g}_1$ maps a component of $\mathbf{m}_1$ to a constant, since $\mathbf{g}_1$ would not be invertible anymore and such a solution would violate the invariance in Eq. (14). To prove a contradiction, suppose that, w.l.o.g., $\mathbf{h}_1(\mathbf{c}, \mathbf{s}, \mathbf{m}_1)_{1:n_c} := \mathbf{h}_1(\mathbf{z}_1)_{1:n_c}$ depends on some component in $\mathbf{m}_1$ in the sense that the partial derivative of $\mathbf{h}_1(\mathbf{z}_1)_{1:n_c}$ w.r.t. some modality-specific variable $m_{1,l}$ is non-zero for some point $(\mathbf{c}^*, \mathbf{s}^*, \mathbf{m}_1^*) \in \mathcal{Z}_1$. Specifically, it implies that the partial derivative $\partial \mathbf{h}_1(\mathbf{z}_1)_{1:n_c} / \partial m_{1,l}$ is positive in a neighborhood around $(\mathbf{c}^*, \mathbf{s}^*, \mathbf{m}_1^*)$, which is a non-empty open set, since $\mathbf{h}_1$ is smooth. On the other hand, due to the independence of $\mathbf{z}_2$ and $\mathbf{m}_1$, the fact that $\mathbf{h}_2(\mathbf{z}_2)_{1:n_c}$ cannot not depend on $\mathbf{m}_1$, and that $p(\mathbf{z}) > 0$ almost everywhere, we come to a contradiction. That is, there exists an open set of points with positive measure, namely the neighbourhood around $(\mathbf{c}^*, \mathbf{s}^*, \mathbf{m}_1^*)$, on which

$$|(\mathbf{h}_1(\mathbf{z}_1)_{1:n_c} - \mathbf{h}_2(\mathbf{z}_2)_{1:n_c})| > 0 \tag{16}$$

almost surely, which contradicts the invariance in Equation (13). The statement does not change, if we add further dependencies of $\mathbf{h}_1$ on components of $\mathbf{m}_1$, or for $\mathbf{h}_2$ on components of $\mathbf{m}_2$, because $\mathbf{m}_1$ and $\mathbf{z}_2$ are independent, and $\mathbf{m}_2$ and $\mathbf{z}_1$ are independent as well. Hence, we show that *any* encoder that minimizes the objective in Equation (5) cannot depend on modality-specific information.

Having established that neither $\mathbf{h}_1(\mathbf{z}_1)_{1:n_c}$, nor $\mathbf{h}_2(\mathbf{z}_2)_{1:n_c}$ can depend on modality-specific information, it remains to show that style information is not encoded, as well. Leveraging Assumption 2, we can show that the strict inequality in Equation (13) has a positve density if $\mathbf{h}_1(\mathbf{z}_1)_{1:n_c}$ or $\mathbf{h}_2(\mathbf{z}_2)_{1:n_c}$ was dependent on a dimension in $\mathbf{s}$ respectively $\tilde{\mathbf{s}}$, which would again lead to a violation of the invariance derived in Equation (13), as shown in von Kügelgen et al. (2021, Proof of Theorem 4.2).

**Step 3.** It remains to show that $\mathbf{h}_1, \mathbf{h}_2$ are bijections. We know that $\mathcal{C}$ and $(0, 1)^{n_c}$ are simply connected and oriented $C^1$ manifolds, and we have established in Step 1 that $\mathbf{h}_1$ and $\mathbf{h}_2$ are smooth and hence differentiable functions. Since $p_\mathbf{c}$ is a regular density, the uniform distributions w.r.t. the pushthrough functions $\mathbf{h}_1$ and $\mathbf{h}_2$ are regular densities. Thus, $\mathbf{h}_1$ and $\mathbf{h}_2$ are bijections (Zimmermann et al., 2021, Proposition 5)

Step 3 concludes the proof. We have shown that for any pair of smooth functions $\mathbf{g}_1, \mathbf{g}_2$ that attain the global minimum of Eq. (5), we have that $\mathbf{c}$ is *block-identified* (Def. 1) by $\mathbf{g}_1$ and $\mathbf{g}_2$. □

### A.2 SYMMETRIC GENERATIVE PROCESS

Throughout the main body of the paper, we described an asymmetric generating mechanism, where $\mathbf{z}_2$ is a perturbed version of $\mathbf{z}_1$. Here, we will briefly sketch out how our model and results can be adapted to a symmetric setting, where *both* $\mathbf{z}_1$ and $\mathbf{z}_2$ are generated as perturbations of $\mathbf{z}$.

Concretely, we would need to make small adjustments to Assumptions 1 and 2 as follows. We start with the content invariance in Assumption 1, which specifies how $\mathbf{z}_1 = (\tilde{\mathbf{c}}_1, \tilde{\mathbf{s}}_1, \tilde{\mathbf{m}}_1)$ and $\mathbf{z}_2 = (\tilde{\mathbf{c}}_2, \tilde{\mathbf{s}}_2, \tilde{\mathbf{m}}_2)$ are generated.

Let $i \in \{1, 2\}$. The conditional density $p_{\mathbf{z}_i|\mathbf{z}}$ over $\mathcal{Z}_i \times \mathcal{Z}$ takes the form

$$p_{\mathbf{z}_i|\mathbf{z}}(\mathbf{z}_i|\mathbf{z}) = \delta(\tilde{\mathbf{c}}_i - \mathbf{c})\delta(\tilde{\mathbf{m}}_i - \mathbf{m}_i)p_{\tilde{\mathbf{s}}_i|\mathbf{s}}(\tilde{\mathbf{s}}_i|\mathbf{s}) , \tag{17}$$

where $\delta(\cdot)$ is the Dirac delta function, i.e., $\tilde{\mathbf{c}}_i = \mathbf{c}$ almost everywhere, as well as $\tilde{\mathbf{m}}_i = \mathbf{m}_i$ almost everywhere. Note that since $\tilde{\mathbf{c}}_1 = \mathbf{c}$ a.e. and $\mathbf{c} = \tilde{\mathbf{c}}_2$ a.e., it follows that $\tilde{\mathbf{c}}_1 = \tilde{\mathbf{c}}_2$ almost everywhere, which is a property that is needed in Step 1 in the proof of Theorem 1. In addition, it still holds

that $\tilde{\mathbf{m}}_i \perp\!\!\!\perp \mathbf{z}_j$, for $i, j \in \{1, 2\}$ and $i \neq j$, which is needed in Step 2 of the proof to show that modality-specific information is not encoded.

Lastly, we need to revisit Assumption 2, for which both $\tilde{\mathbf{s}}_1$ and $\tilde{\mathbf{s}}_2$ would be generated through perturbations of $\mathbf{s}$ via the conditional distribution $p_{\tilde{\mathbf{s}}_i|\mathbf{s}}$ on $\mathcal{S} \times \mathcal{S}$, as described in Assumption 2, for each $i$ individually. As a small technical nuance, we would need to specify the conditional generation of the perturbed style variables $\tilde{\mathbf{s}}_1$ and $\tilde{\mathbf{s}}_2$ such that they are not perturbed in an identical manner w.r.t. $\mathbf{s}$. This can be ensured by, e.g., constraining $p_A$ appropriately to exclude the degenerate case where dimensions in $\tilde{\mathbf{s}}_1$ and $\tilde{\mathbf{s}}_2$ are perfectly aligned—a case that needs to be excluded for Step 2 of the proof of Theorem 1.

## B EXPERIMENTS

### B.1 EXPERIMENTAL DETAILS

**Numerical simulation** The generative process is described in Section 5.1. Here, we provide additional information about the experiment. The invertible MLP is constructed similar to previous work (Hyvärinen and Morioka, 2016; Hyvärinen and Morioka, 2017; Zimmermann et al., 2021; von Kügelgen et al., 2021) by resampling square weight matrices until their condition number surpasses a threshold value. For the original setting ($\mathbf{f}_1 = \mathbf{f}_2$), we use one encoder ($\mathbf{g}_1 = \mathbf{g}_2$), whereas for the multimodal setting ($\mathbf{f}_1 \neq \mathbf{f}_2$), we use distinct encoders ($\mathbf{g}_1 \neq \mathbf{g}_2$) to mirror the assumption of distinct mixing functions and because, in practice, the dimensionality of the observations can differ across modalities. In Table 2a, we specify the main hyperparameters for the numerical simulation.

**Multimodal3DIdent** Our dataset of image/text pairs is based on the code used to generate the *Causal3DIdent* (von Kügelgen et al., 2021; Zimmermann et al., 2021) and *CLEVR* (Johnson et al., 2017) datasets. Images are generated using the *Blender* renderer (Blender Online Community, 2018). The rendering serves as a complex mixing function that generates the images from 11 different parameters (i.e., latent factors) that are listed in Table 3. To generate textual descriptions, we adapt the text rendering from *CLEVR* (Johnson et al., 2017) and use 5 different phrases to induce modality-specific variation. The latent factors used to generate the text are also listed in Table 3. The dependence between the image and text modality is determined by three content factors (object shape, x-position, and y-position) and one style factor (object color). For the object color in the image, we use a continuous hue value, whereas for the text we match the RGB value with the nearest color value from one of three different palettes[7] that is sampled uniformly at random for each observation. Further, we ensure that there are no overlapping color values across palettes by using a prefix for the respective palette (e.g., "xkcd:black") when necessary. In Section 5.2, we use a version of the *Multimodal3DIdent* dataset with a causal dependence from content to style. Specifically, the color of the object depends on its x-position. In particular, we split the range of hue values $[0, 1]$ into three equally sized intervals and associate each of these intervals with a fixed x-position of the object. For instance, if x-position is "left", we sample the hue value from the interval $[0, 1/3]$. Consequently, the color of the object can be predicted to some degree from the position of the object. Samples of image/text pairs from the *Multimodal3DIdent* dataset are shown in Figures 2 and 4. The hyperparameters for the experiment are listed in Table 2b. In Appendix B.2, we provide additional results for a version of the dataset with mutually independent factors.

**High-dimensional image pairs** In Appendix B.2, we provide additional results using a dataset of high-dimensional pairs of images of size 224x224x3. Similar to *Multimodal3DIdent*, images are generated using *Blender* (Blender Online Community, 2018) and code adapted from previous work (Zimmermann et al., 2021; von Kügelgen et al., 2021). Each image depicts a scene with one type of object (a teapot, like in Zimmermann et al., 2021) in front of a colored background and illuminated by a colored spotlight (for examples, see Figure 9). The scene is defined by 9 continuous latent variables each of which is sampled from a uniform distribution. Object positions (x-, y- and z-coordinates) are content factors that are always shared between modalities, while object-, spotlight- and background-colors are style factors that are stochastically shared. Modality-specific factors are object rotation ($\alpha$ and $\beta$ angles) for one modality and spotlight position for the other. To simulate modality-specific mixing functions, we render the objects using distinct textures (i.e., rubber and metallic) for each modality. Further, we generate two versions of this dataset, with and without

causal dependencies. For the dataset with causal dependencies we sample the latent factors according to a causal model, where background-color depends on z-position and spotlight-color depends on object-color. We use ResNet-18 encoders and similar hyperparameter values to those used for the image/text experiment (Table 2b).

| Parameter | Value |
|---|---|
| Generating function | 3-layer MLP |
| Encoder | 7-layer MLP |
| Optimizer | Adam |
| Cond. threshold ratio | 1e-3 |
| Dimensionality $d$ | 15 |
| Batch size | 6144 |
| Learning rate | 1e-4 |
| Temperature $\tau$ | 1.0 |
| # Seeds | 3 |
| # Iterations | 300,000 |
| Similarity metric | Euclidian |
| Gradient clipping | 2-norm; max value 2 |

(a) Parameters used for the numerical simulation.

| Parameter | Value |
|---|---|
| Generating function | Image and text rendering |
| Image encoder | ResNet-18 |
| Text encoder | 4-layer ConvNet |
| Optimizer | Adam |
| Batch size | 256 |
| Learning rate | 1e-5 |
| Temperature $\tau$ | 1.0 |
| # Seeds | 3 |
| # Iterations | 100,000 |
| # Samples (train / val / test) | 125,000 / 10,000 / 10,000 |
| Similarity metric | Cosine similarity |
| Gradient clipping | 2-norm; max value 2 |

(b) Parameters used for *Multimodal3DIdent*.

Table 2: Experimental parameters and hyperparameters used for the two experiments in the main text.

| Latent factor | Distribution | Details |
|---|---|---|
| **Object shape** | Categorical | 7 unique values |
| **Object x-position** | Categorical | 3 unique values |
| **Object y-position** | Categorical | 3 unique values |
| *Object color* | Uniform | hue value in $[0, 1]$ |
| Object rotation $\alpha$ | Uniform | angle value in $[0, 1]$ |
| Object rotation $\beta$ | Uniform | angle value in $[0, 1]$ |
| Object rotation $\gamma$ | Uniform | angle value in $[0, 1]$ |
| Spotlight position | Uniform | angle value in $[0, 1]$ |
| Spotlight color | Uniform | hue value in $[0, 1]$ |
| Background color | Uniform | hue value in $[0, 1]$ |
| **Object shape** | Categorical | 7 unique values |
| **Object x-position** | Categorical | 3 unique values |
| **Object y-position** | Categorical | 3 unique values |
| *Object color* | Categorical | color names (3 palettes)[7] |
| Text phrasing | Categorical | 5 unique values |

Table 3: Description of the latent factors used to generate *Multimodal3DIdent*. The first 10 factors are used to generate the images and the remaining 5 factors are used to generate the text. Object z-position is kept constant for all images, which is why we do not list it among the generative factors. Independent factors are drawn uniformly from the respective distribution. Content factors are denoted in bold and style factors in italic; the remaining factors are modality-specific.

---

[7] We use the following three palettes from the `matplotlib.colors` API: Tableau colors (10 values), CSS4 colors (148 values), and XKCD colors (949 values).

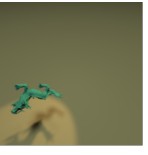 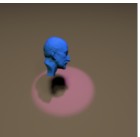 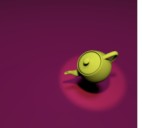 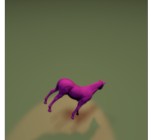 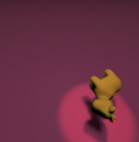 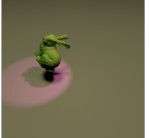 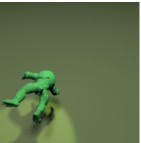

| On the **bottom-left** of the image there is a **tab:cyan** object in the form of a **dragon**. | On the **top-center** of the image there is a **tab:cyan** object in the form of a **head**. | A **green yellow teapot** is on the **mid-right** of the image. | An **horse** of **tab:pink color** is visible positioned in the **bottom-center** of the image. | A **lawn green cow** is on the **bottom-right** of the image. | A **xkcd:poison green hare** is on the **top-left** of the image. | A **spring green armadillo** is on the **bottom-left** of the image. |

Figure 4: Examples of image/text pairs from the *Multimodal3DIdent* dataset. Each sample shows one of the seven shapes or classes of objects included in the dataset.

| Generative process | | | $R^2$ (nonlinear) | | | Generative process | | | $R^2$ (nonlinear) | |
|---|---|---|---|---|---|---|---|---|---|---|
| p(chg.) | Stat. | Cau. | Content c | Style s | | p(chg.) | Stat. | Cau. | Content c | Style s |
| 1.0 | ✗ | ✗ | **1.00** ± 0.00 | 0.00 ± 0.00 | | 1.0 | ✗ | ✗ | **1.00** ± 0.00 | 0.00 ± 0.00 |
| 0.75 | ✗ | ✗ | **0.99** ± 0.01 | 0.00 ± 0.00 | | 0.75 | ✗ | ✗ | **1.00** ± 0.00 | 0.00 ± 0.00 |
| 0.75 | ✓ | ✗ | **0.99** ± 0.00 | 0.52 ± 0.09 | | 0.75 | ✓ | ✗ | **0.99** ± 0.01 | 0.36 ± 0.10 |
| 0.75 | ✗ | ✓ | **1.00** ± 0.00 | **0.79** ± 0.04 | | 0.75 | ✗ | ✓ | **1.00** ± 0.00 | **0.81** ± 0.03 |
| 0.75 | ✓ | ✓ | **0.99** ± 0.01 | **0.81** ± 0.04 | | 0.75 | ✓ | ✓ | **0.99** ± 0.01 | **0.83** ± 0.05 |

(a) Original setting  (b) Multimodal setting

Table 4: Results of the numerical simulations *without* modality-specific latent variables. We compare the original setting ($\mathbf{f}_1 = \mathbf{f}_2$, left table) with the multimodal setting ($\mathbf{f}_1 \neq \mathbf{f}_2$, right table). Each row presents the results of a different setup with varying style-change probability p(chg.) and possible statistical (Stat.) and/or causal (Caus.) dependencies. Each value denotes the $R^2$ coefficient of determination (averaged across 3 seeds) for a nonlinear regression model that predicts the respective ground truth factor ($\mathbf{c}$, $\mathbf{s}$, or $\mathbf{m}_i$) from the learned representation.

## B.2 Additional experimental results

**Numerical simulation without modality-specific latents**    Recall that the considered generative process (Section 3) has two sources of modality-specific variation: modality-specific mixing functions and modality-specific latent variables. To decouple the effect of these two sources of variation, we conduct an ablation study *without* modality-specific latent variables. Table 4 presents the results, showing that content is block-identified in both the original setting ($\mathbf{f}_1 = \mathbf{f}_2$, Table 4a) and the multimodal setting ($\mathbf{f}_1 \neq \mathbf{f}_2$, Table 4b). Compared to Table 1, we observe that the content prediction is improved slightly in the case without modality-specific latent variables. Hence, our results suggest that contrastive learning can block-identify content factors in the multimodal setting with and without modality-specific latent variables.

**Numerical simulation with discrete latent factors**    Extending the numerical simulation from Section 5.1, we test block-identifiability of content information when observations are generated from a mixture of continuous and discrete latent variables, thus violating one of the assumptions from Theorem 1. In this setting, content, style and modality-specific information are random variables with 5 components sampled from either a continuous normal distribution or a discrete multinomial distribution with $k$ classes, for which we experiment with different $k \in \{3, 4, \ldots, 10\}$. For all settings, we train an encoder with the InfoNCE objective and set the encoding size to 5 dimensions. The other hyperparameters used in this set of experiments are detailed in Table 2a. To ensure convergence of the models, we extended the number of training iterations to 600,000 and 3,000,000 for experiments with discrete style/modality-specific and discrete content variables respectively. With discrete style or modality-specific variables and continuous content (Figures 5a and 5b), the results suggest that content is block-identified, since the prediction of style and modality-specific information is at chance level (i.e., *accuracy* = $1/k$) while content is consistently fully recovered ($R^2 \geq 0.99$). In the opposite setting, with continuous style and modality-specific variables and discrete content (Figure 5c), the number of content classes appears to be a critical factor for block-identifiability of content: while content is always encoded well, style information is also encoded to a significant extent when the number of content classes is small, but significantly less style can be recovered when the number

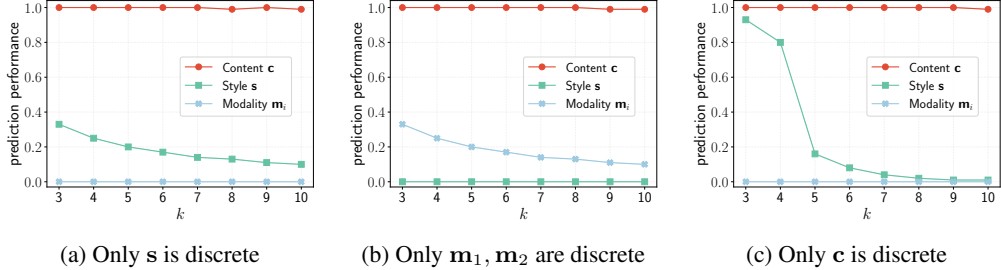

(a) Only $\mathbf{s}$ is discrete       (b) Only $\mathbf{m}_1, \mathbf{m}_2$ are discrete       (c) Only $\mathbf{c}$ is discrete

Figure 5: Numerical simulations with discrete latent factors. The results show three settings in each of which one group of latent variables is discrete while the remaining groups are continuous. Continuous variables are normally distributed, whereas discrete variables are sampled from a multinomial distribution with $k$ distinct classes. We measure the prediction performance with a nonlinear model in terms of the $R^2$ coefficient of determination for continuous factors and classification accuracy for discrete factors respectively. Each point denotes the average across three seeds and error bars show the standard deviation, which is relatively small.

of content classes increases. Through this set of experiments, we challenge the assumption that *all* generative factors should be continuous (c.f., Section 3) and show that block-identifiability of content can still be satisfied when content is continuous while style or modality-specific variables are discrete. On the other hand, style is encoded to a significant extent when content is discrete, which might explain our observation for the image/text experiment, where we saw that, in the presence of discrete content factors, some style information can be encoded.

**Dimensionality ablations for the numerical simulation** To test the effect of latent dimensionality on identifiability, Figure 6 presents dimensionality ablations where we keep the number of content dimensions fixed and only vary the number of style or modality-specific dimensions, $n_s$ and $n_m$ respectively. Figures 6a and 6c confirm that block-identifiability of content still holds when we significantly increase the number of style or modality-specific dimensions, as the representation consistently encodes only content and no style or modality-specific information. In Figures 6b and 6d, we can observe that the training loss decreases more slowly when we increase the dimensionality of $n_c$ and $n_s$ respectively, which provides an intuition that the sample complexity might increase with the number of style and modality-specific dimensions.

**Estimating the number of content factors** The estimation of the number of content factors is an important puzzle piece, since Theorem 1 assumes that the number of content factors is known or that it can be estimated. In practice, the number of content factors can be viewed as a single hyperparameter (e.g., Locatello et al., 2020) that can be tuned with respect to a suitable model selection metric. For instance, one could use the validation loss for model selection, which would be convenient since the validation loss only requires a holdout dataset and no additional supervision. In Figure 7, we plot the validation loss (averaged over 2,000 validation samples) as a function of the encoding size for both experiments used in our paper. Results for the numerical simulation are shown in Figure 7a and for the image/text experiment in Figure 7b. For both datasets, we observe that the validation loss increases most significantly in the range around the true number of content factors. For the numerical simulation, the results look promising as they show a clear "elbow" (James et al., 2013) at the correct value of 5, which corresponds to the true number of content factors. The results are less clear for the image/text experiment, where the elbow method might suggest the range of 2-4 content factors, while the true value is 3. While these initial results look promising, we believe that more work is required to investigate the estimation of the number of content factors and the design of suitable heuristics, which are interesting directions for future research.

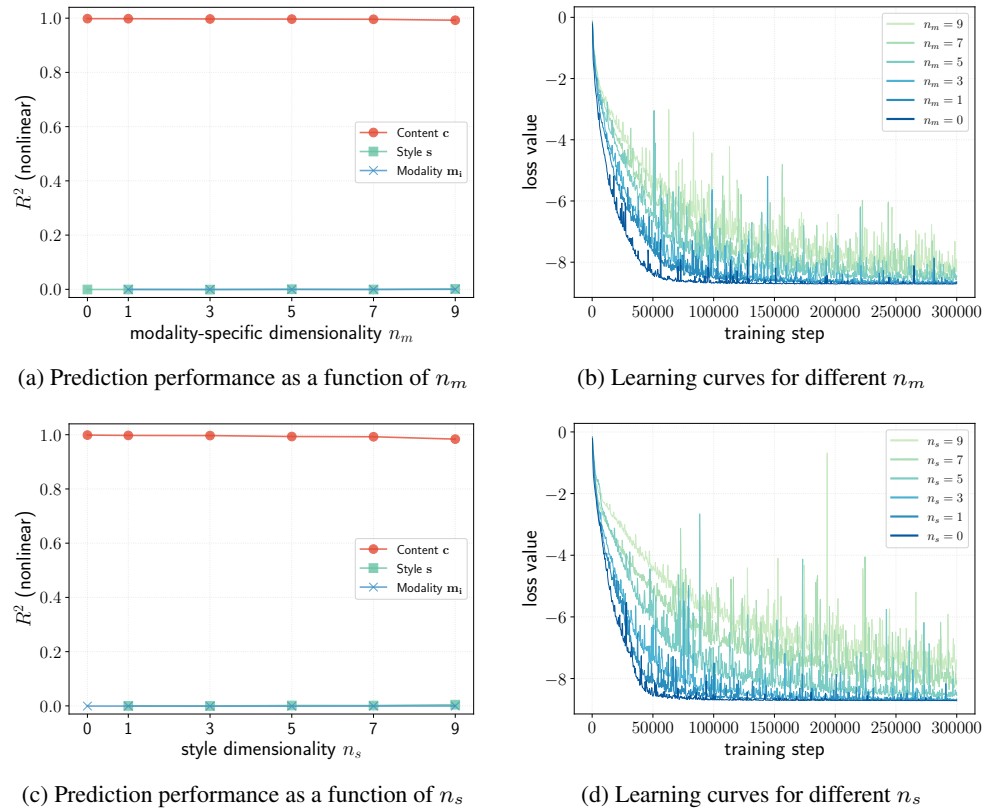

(a) Prediction performance as a function of $n_m$

(b) Learning curves for different $n_m$

(c) Prediction performance as a function of $n_s$

(d) Learning curves for different $n_s$

Figure 6: Dimensionality ablation for the numerical simulation. We consider the multimodal setting with mutually independent factors and test the effect of latent dimensionality on identifiability by keeping the number of content dimensions fixed and only varying the number of style or modality-specific dimensions ($n_s$ and $n_m$ respectively). In Figures 6a and 6c we measure the nonlinear prediction performance in terms of the $R^2$ coefficient of determination of a nonlinear regression model that predicts the respective ground truth factor ($\mathbf{c}$, $\mathbf{s}$, or $\mathbf{m}_i$) from the learned representation. In Figures 6b and 6d, we plot the learning curves (i.e., the training loss) of the respective models to compare how fast they converge.

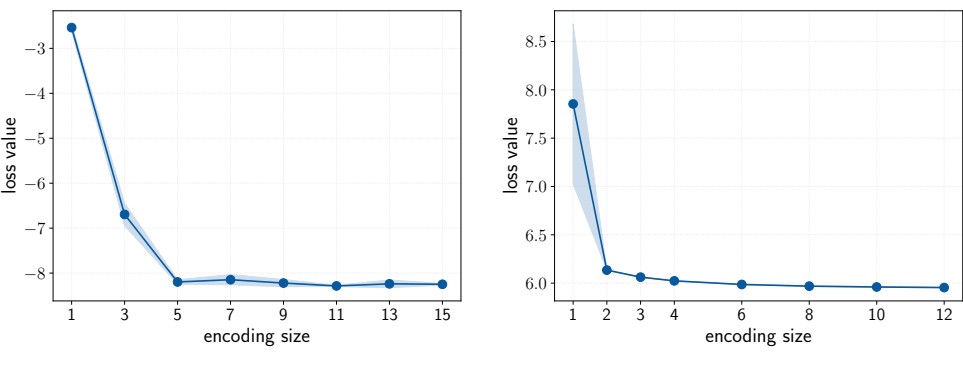

(a) Validation loss for the numerical simulation

(b) Validation loss for the image/text experiment

Figure 7: An attempt at estimating of the number of content factors using the validation loss. The validation loss corresponds to the value of the $\mathcal{L}_{\mathrm{SymInfoNCE}}$ objective computed on a holdout dataset. Since we are interested in estimating the true number of content factors to select the encoding size appropriately, we plot the validation loss as a function of the encoding size. We show the validation loss for the numerical simulation with independent factors (Figure 7a) and for the image/text experiment (Figure 7b) respectively.

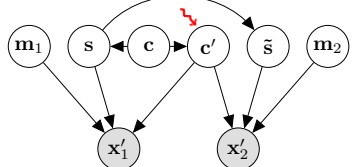

| Generative process | | | $R^2$ (nonlinear) | | | |
|---|---|---|---|---|---|---|
| p(chg.) | Stat. | Cau. | Content c | Content c$'$ | Style s | Modality $m_i$ |
| 1.0 | ✗ | ✗ | $0.00 \pm 0.00$ | $\mathbf{1.00} \pm 0.00$ | $0.00 \pm 0.00$ | $0.00 \pm 0.00$ |
| 0.75 | ✗ | ✗ | $0.00 \pm 0.00$ | $\mathbf{1.00} \pm 0.00$ | $0.00 \pm 0.00$ | $0.00 \pm 0.00$ |
| 0.75 | ✓ | ✗ | $0.00 \pm 0.00$ | $\mathbf{1.00} \pm 0.00$ | $\underline{0.50} \pm 0.19$ | $0.00 \pm 0.00$ |
| 0.75 | ✗ | ✓ | $0.01 \pm 0.00$ | $\mathbf{0.98} \pm 0.00$ | $0.03 \pm 0.01$ | $0.00 \pm 0.00$ |
| 0.75 | ✓ | ✓ | $\underline{0.28} \pm 0.14$ | $\mathbf{0.91} \pm 0.03$ | $\underline{0.39} \pm 0.20$ | $0.00 \pm 0.00$ |

Figure 8: Evaluation with test-time interventions. We use the interventional setup that is illustrated on the left, i.e., perturbed samples $\mathbf{x}_1', \mathbf{x}_2'$ that are generated from the intervened content $\mathbf{c}'$, which is a copy of the original content $\mathbf{c}$ with an intervention, i.e., a batch-wise permutation ($\rightsquigarrow$) that makes $\mathbf{c}'$ independent of $\mathbf{s}$. Each row presents the results of a different setup with varying style-change probability p(chg.) and possible statistical (Stat.) and/or causal (Caus.) dependencies. Each value denotes the $R^2$ coefficient of determination (averaged across 3 seeds) for a nonlinear regression model that predicts the respective ground truth factor ($\mathbf{c}, \mathbf{c}', \mathbf{s}$, or $\mathbf{m}_i$) from the learned representation.

**Evaluation with test-time interventions**   Previously, we observed that style can be predicted to some degree when there are causal dependencies from content to style (Table 1), which can be attributed to style information being partially predictable from the encoded content information in the causal setup. To verify that the encoders only depend on content information (i.e., that content is block-identified), we assess the trained models using a novel, more rigorous empirical evaluation for the numerical simulation. We test the effect of *interventions* $\mathbf{c} \rightarrow \mathbf{c}'$, which perturb the content information *at test time* via batch-wise permutations of content, before generating $\mathbf{x}_1' = \mathbf{f}_1(\mathbf{c}', \mathbf{s}, \mathbf{m}_1)$ and $\mathbf{x}_2' = \mathbf{f}_1(\mathbf{c}', \tilde{\mathbf{s}}, \mathbf{m}_1)$. Hence, we break the causal dependence between content and style (see illustration in Figure 8), which allows us to better assess whether the trained encoders depend on content or style information. Specifically, we train the encoders for 3,000,000 iterations to ensure convergence and then train nonlinear regression models to predict both the original and the intervened content variables from the learned representations. Figure 8 presents our results using the interventional setup, showing that in most cases only content information can be recovered. We observe an exception (underlined values) in the two cases with statistical dependencies, where some style information can be recovered, which is expected because statistical dependencies reduce the effective dimensionality of content (cp. von Kügelgen et al., 2021). Analogously, in the case of statistical and causal dependencies, some of the original content information can be recovered via the encoded style information. In summary, the evaluation with interventions provides a more rigorous assessment of block-identifiability in the causal setup, showing that neither style nor modality-specific information can be recovered when the encoding size matches the true number of content dimensions.

**High-dimensional image pairs with continuous latents**   To bridge the gap between continuous and discrete data, we provide an additional experiment that offers a realistic setup but uses only continuous latent variables to satisfy the assumptions of Theorem 1. Previously, in Section 5.2, we used a complex multimodal dataset of image/text pairs, which were generated from a combination of continuous and discrete latent factors. Now, we consider a different dataset that consists of *pairs high-dimensional images* generated from a set of continuous latents, which is more in line with our theoretical assumptions. Note that datasets with pairs of images are common in practice, for example, in medical imaging where patients are assessed using multiple views

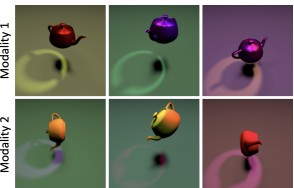

Figure 9: Examples of high-dimensional image pairs.

(e.g., images from different angles) or multiple modalities (e.g., as in PET-CT imaging). To generate the data, we adapt the code from 3DIdent (Zimmermann et al., 2021) to render pairs of images, for which the object position is always shared (i.e., content), the object-, spotlight- and background-color is stochastically shared (i.e., style), and modality-specific factors are object rotation for one modality and spotlight position for the other. Additionally, we render the objects using different textures to simulate a modality-specific mixing process. Samples of image pairs are shown in Figure 9 and further details about the dataset can be found in Appendix B.1. We train the encoders with the InfoNCE objective for 60,000 iterations using the same architectures and hyperparameters as for *Multimodal3DIdent* (Table 2b), and again evaluate the $R^2$ coefficient of determination using a kernel ridge regression that predicts the respective ground truth factor from the learned representations.

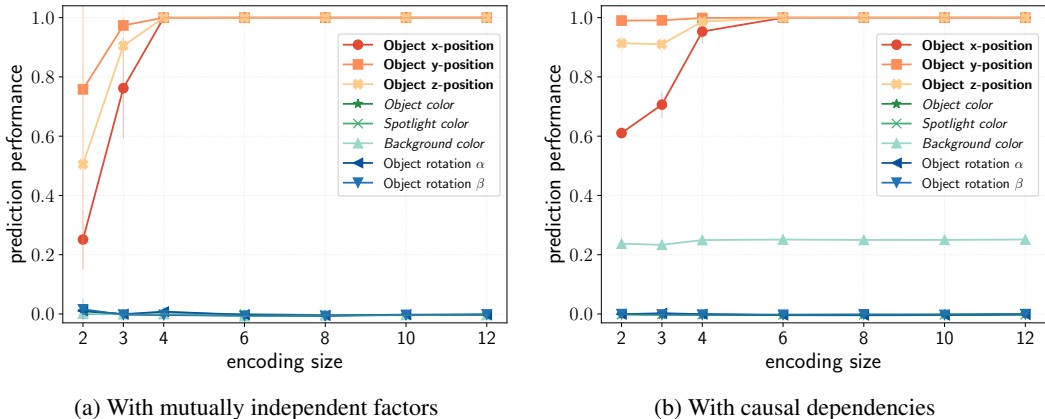

(a) With mutually independent factors

(b) With causal dependencies

Figure 10: Result with pairs of high-dimensional images. As a function of the encoding size of the model, we assess the nonlinear prediction of ground truth factors to quantify how well the learned representation encodes the respective factors. Content factors are denoted in bold, style factors in italic, and modality-specific factors in regular font. Each point denotes the average $R^2$ score across three seeds and bands show one standard deviation.

Figure 10 present our results for the dataset of image pairs, showing the prediction performance as a function of the encoding size for the setting with causal dependencies (Figure 10b) and the setting with mutually independent latent variables (Figure 10a) respectively. In both settings, content information (i.e. object position) is recovered when sufficient encoding capacity is available. Style and modality-specific information, on the other hand, are discarded independent of the encoding size. In Figure 10b we observe the recovery of some style information, which is expected because style can be predicted to some degree from the encoded content information when there is a causal dependence of style on content. Overall, these findings lend further support to our theoretical result from Theorem 1, as we investigate a realistic setting with only continuous latent factors, which is more in line with our assumptions. Notably, the results appear more consistent with our theory, e.g., showing that less style and modality-specific information is encoded, compared to our results for the image/text experiment, where we used a combination of continuous and discrete latent factors.

**Multimodal3DIdent with mutually independent factors**  For the results of the image/text experiment in the main text (Section 5.2) we used the *Multimodal3DIdent* dataset, which we designed such that object color is causally dependent on the x-position of the object to impose a causal dependence of style on content. In Figure 11, we provide a similar analysis using a version of the dataset *without* the causal dependence, i.e., with mutually independent factors. For both modalities, we observe that object color is only encoded when the encoding size is larger than four, i.e., when there is excess capacity beyond the capacity needed to encode all content factors. Hence, these results corroborate that contrastive learning can block-identify content factors in a complex multimodal setting with heterogeneous image/text pairs.

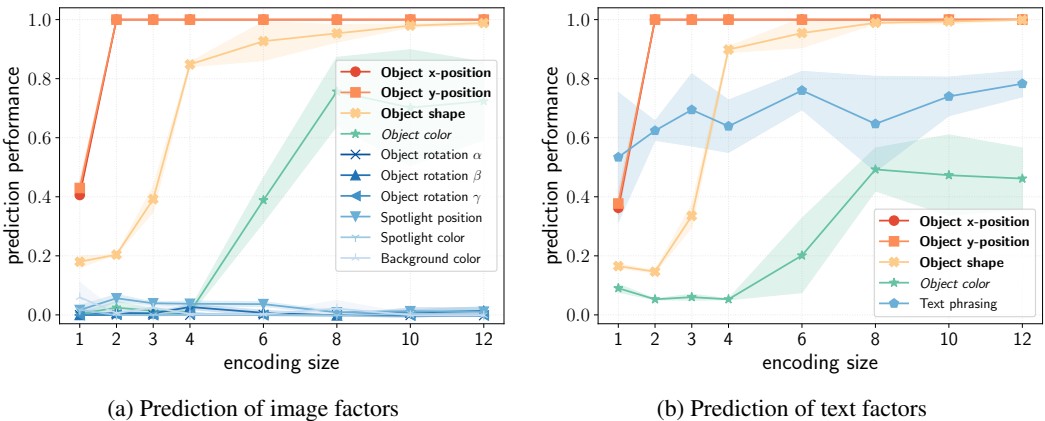

(a) Prediction of image factors

(b) Prediction of text factors

Figure 11: Result on *Multimodal3DIdent* with mutually independent factors. As a function of the encoding size of the model, we assess the nonlinear prediction of ground truth image factors (left subplot) and text factors (right subplot) to quantify how well the learned representation encodes the respective factors. Content factors are denoted in bold and style factors in italic. Along the x-axis, we vary the encoding size, i.e., the output dimensionality of the model. We measure the prediction performance in terms of the $R^2$ coefficient of determination for continuous factors and classification accuracy for discrete factors respectively. Each point denotes the average across three seeds and bands show one standard deviation.

