# OpenReview forum: "Identifiability Results for Multimodal Contrastive Learning"
_ICLR.cc/2023/Conference — ICLR 2023 poster_

### Official Review · Reviewer_YHeS · 2022-10-22

**Confidence:** 5
**Correctness:** 4
**Technical Novelty And Significance:** 4
**Empirical Novelty And Significance:** 3
**Recommendation:** 8

**Clarity, Quality, Novelty And Reproducibility:**

Clarity:
- As I said, this paper is very clearly written.
- Assumption 2: I was a little bit confused at first with how (8) relates to p_(\tilde{s} | s). It might be useful to write explicitly that p(\tilde{s} | s) = sum_A p(A)p(\tilde{s} | s, A).
- Definition 1: “...contains all and only information about c, i.e., if there exists an invertible function…”. What is said before the “i.e.” is different from what is said after it: even if \hat{c} = h(c) for some invertible function h, \hat{c} might still contain information about s. Statistical independence and disentanglement are not the same. This is actually discussed in 5.1. A similar problematic formulation is employed in “Implication and scope” of Section 7.
- Theorem 1: I was slightly confused when reading that p_{z} must be a continuous density, since at first I thought z contains \tilde{s}, and if that were the case, that would imply that p(A) puts all of its mass on A = {1,..., n_s}. But z does not contain \tilde{s}. Maybe that’s worth emphasizing.

Novelty/originality:
- I have never seen this result before.
- Even if this work clearly builds on previous identifiability results, it seems to extend them in a non-trivial way.

Improvement suggestions:
- I could not find the full joint over p(z) in the main text. It might be worth writing it down somewhere, to make explicit the fact that s can depend on c.
- Looking at the generative model for the latent variable z (figure 1) I find myself wondering which pieces of the model are absolutely crucial for the identifiability result to follow through and which are just there for sake of generality. For example, does the result follow through when either m_1 and m_2 are empty (i.e. just absent from the model)? Is dependency between s and c necessary? Is the dependency between s and \tilde{s} necessary? What if s and \tilde{s} are empty?

Minor:
- Typo in Assumption 2 (ii). “... is smooth w.r.t. Both \tilde{s}_A and s_A” ?
- The authors might want to consider citing “Disentanglement via Mechanism Sparsity Regularization: A New Principle for Nonlinear ICA” by Lachapelle et al (2022) in “Limitation and outlook” when mentioning interventions on the generative process and actions in reinforcement learning.

**Strength And Weaknesses:**

Strengths:
- This paper is very well written and fun to read (even on a Saturday morning).
- The theoretical contribution is significant.
- The technical assumptions are very clearly presented, a quality which is sometimes lacking in this line of work.
- The topic is timely, given how popular multimodal contrastive learning is.
- Presents a realistic experiments with image/text data.

Weaknesses:
- I would have appreciated more experiments where theoretical assumptions are violated, for example, what happens if Assumption 2 doesn't hold?
- The theory applies only to continuous data (as is made clear by the assumption that both encoders f_1 and f_2 are diffeomorphisms), but the main applications of multimodal contrastive learning is with image/text data.

**Summary Of The Paper:**

This work proposes a novel identifiability analysis of a certain class of multimodal contrastive learning algorithms. The result is centered around a model with a latent variable partitioned into content (shared across modalities), style and and modality-specific information. The main theoretical result guarantees a form of block identifiability of the shared latent content. This work distinguishes itself by allowing for certain types of dependencies between the latent factors and by allowing for two distinct decoders, corresponding to the two modalities. The theory is illustrated in both a controlled synthetic dataset and a more realistic one presenting image/text pairs.

**Summary Of The Review:**

I very much enjoyed reading this manuscript. It is well written, easy to follow and its topic is very timely. The identifiability results are novel and very clearly presented. I found the experiments with the realistic image/text data to be very convincing, although I was expecting more experiments where the assumptions of the theory are violated. The main drawback is that the theory applies only to continuous data and that one of the main application of multimodal contrastive learning is image/text data, but this point is acknowledge by the authors. Still, this remains an important contribution and I am very happy to recommend acceptance.

---

> ### Author Response · Authors · 2022-11-19
> **Response to Reviewer YHeS (part 2/2)**
>
> > Definition 1: “...contains all and only information about c, i.e., if there exists an invertible function…”. What is said before the “i.e.” is different from what is said after it: even if $\hat{c} = h(c)$ for some invertible function h, $\hat{c}$ might still contain information about s. Statistical independence and disentanglement are not the same. This is actually discussed in 5.1. A similar problematic formulation is employed in “Implication and scope” of Section 7.
>
> Thank you for your remark. We understand that this phrase can be confusing and therefore adapted the definition to include only the mathematically precise statement. What we mean, when we write "the encoder depends on/encodes all and only content", is that the gradients of the encoders $g_1$ and $g_2$ are non-zero only w.r.t. the content dimensions (see proof of Theorem 1, Step 2). As you have suggested, we also adapted the phrasing in Section 7 to be more precise.
>
> You correctly point out that the empirical results in Section 5.1 include a case where style depends on content, and therefore style information can be predicted from the learned representation even if content is block-identified. To verify that the encoders only depend on content information, in Appendix B.2 [Figure 8](https://ibb.co/M7h0gHD) we provide a novel, more rigorous, empirical evaluation for the numerical simulation. We test the effect of *interventions* $c \to c'$, which perturb the content information before generating $x_1' = f_1(c', s, m_1)$ and $x_2' = f_1(c', \tilde{s}, m_2)$ to break the causal dependence between content and style *at test time*. We observe that---even in the causal setting---only the intervened content, and neither style nor modality-specific information, can be recovered, which supports block-identifiability of content.
>
>
> > Theorem 1: I was slightly confused when reading that p_{z} must be a continuous density, since at first I thought z contains \tilde{s}, and if that were the case, that would imply that p(A) puts all of its mass on A = {1,..., n_s}. But z does not contain \tilde{s}. Maybe that’s worth emphasizing.
>
> > I could not find the full joint over p(z) in the main text. It might be worth writing it down somewhere, to make explicit the fact that s can depend on c.
>
> Thank you for the helpful suggestions. We clarified both points in Section 3.1 (before the last paragraph).
>
> > which pieces of the model are absolutely crucial for the identifiability result to follow through and which are just there for sake of generality. For example, does the result follow through when either m_1 and m_2 are empty (i.e. just absent from the model)? Is dependency between s and c necessary? Is the dependency between s and \tilde{s} necessary? What if s and \tilde{s} are empty?
>
> The essential part is that there are invariant factors (i.e., content $c$) that can be recovered from both modalities (i.e., invertible mixing functions). The result would follow through without modality-specific information $m_1, m_2$ and without stochastically shared factors $s$, $\tilde{s}$. For example, if we would remove the dependence from $s$ on $\tilde{s}$, the latter would become modality-specific, i.e., similar to $m_2$. If, in addition, we would remove the edge between $c$ and $s$, the latter would also become modality-specific, i.e., similar to $m_1$. Then, our assumptions would be closer to the classical setting of multi-view nonlinear ICA, in which all latents are assumed to be mutually independent (e.g., Gresele et al., 2019). Our experiments also include settings with mutually independent latents, for example, in Table 1 (first two rows) and in the new [Figure 10a](https://ibb.co/h9rBZnf)).
>
> > Minor Points
>
> We greatly appreciate these points! We included your suggestions in the updated manuscript.
>
> > I was expecting more experiments where the assumptions of the theory are violated. The main drawback is that the theory applies only to continuous data and that one of the main application of multimodal contrastive learning is image/text data, but this point is acknowledge by the authors. Still, this remains an important contribution and I am very happy to recommend acceptance.
>
> Thank you for your valuable feedback. We hope that our response and new experiments answer your remaining questions, specifically, with regard to the violations of the assumptions and the gap between continuous and discrete data.

---

> > ### Comment · Reviewer_YHeS · 2022-11-23
> > **Response**
> >
> > I thank the authors for their careful consideration of the points raised in my review. I'm satisfied with their answers. I read the other reviews and decided to maintain my evaluation.

---

> > > ### Author Response · Authors · 2022-11-24
> > > **Thank you for your feedback!**
> > >
> > > Thank you for your feedback and the careful consideration of the rebuttal and all other reviews. We are delighted to see that our responses addressed your concerns and greatly appreciate your valuable feedback.

---

> ### Author Response · Authors · 2022-11-19
> **Response to Reviewer YHeS (part 1/2)**
>
> > I very much enjoyed reading this manuscript. It is well written, easy to follow and its topic is very timely. The identifiability results are novel and very clearly presented. I found the experiments with the realistic image/text data to be very convincing, although I was expecting more experiments where the assumptions of the theory are violated. The main drawback is that the theory applies only to continuous data and that one of the main application of multimodal contrastive learning is image/text data, but this point is acknowledge by the authors. Still, this remains an important contribution and I am very happy to recommend acceptance.
>
> Thank you for the positive and very constructive feedback.
>
> In the following, we address your questions regarding further violations and provide new experiments to bridge the gap between continuous and discrete data, as well as a more rigorous evaluation of block-identifiability in the causal setting.
>
> > I would have appreciated more experiments where theoretical assumptions are violated, for example, what happens if Assumption 2 doesn't hold?
>
> Thank you for the excellent question. Assumption 2 is rather general because it allows for style changes that *jointly* perturb subsets of style variables (i.e., not only dimension-wise perturbations); however, we can conceive of two interesting special cases. In one extreme case, when a variable (say $s_l$) is always perturbed in such a way that $\tilde{s}_l$ becomes independent of $s_l$, the perturbed variable would fall into the category of modality-specific information. On the other side of the spectrum, if the probability of perturbation of $\tilde{s}_l$ is zero, the variable would fall into the category of content as it would always be shared between modalities. Therefore, violations of Assumption 2 blur the line between content, style, and modality-specific information, and hence we can relate these violations to cases that we already investigate. This is why we think it would be an interesting research direction to reframe the generative process in terms of *only* stochastically shared variables (with additional causal dependencies) and to study identifiability in this context, as we point out in Section 7.
>
>
> > The theory applies only to continuous data (as is made clear by the assumption that both encoders f_1 and f_2 are diffeomorphisms), but the main applications of multimodal contrastive learning is with image/text data.
>
> To bridge the gap between continuous and discrete data, we provide two new experiments in Appendix B.2: first, an extension of the numerical simulation with discrete latents ([Figure 5](https://ibb.co/ZYMGgXF)); second, a more realistic application with pairs of high-dimensional images generated from continuous latents ([Figure 10](https://ibb.co/h9rBZnf)). In Figure 5a, we observe that block-identifiability of content is still satisfied when content is continuous and style is discrete, while in the setting of discrete content and continuous style (Figure 5b), the number of content classes can be a critical factor for block-identifiability. Second, for the application with high-dimensional image pairs that are generated from continuous latents, Figure 10 shows that block-identifiability is still satisfied in a setting that is more similar to the theoretical setup than the application with image/text data; we even observe that significantly less style and modality-specific information is encoded compared to the image/text results.
>
> > Assumption 2: I was a little bit confused at first with how (8) relates to $p(\tilde{s} | s)$. It might be useful to write explicitly that $p(\tilde{s} | s) = \sum_A p(A)p(\tilde{s} | s, A)$.
>
> Thank you for the suggestion! Indeed, we think it’s helpful to denote this explicitly and now write out $p(\tilde{s} | s) = \sum_A p(A)p(\tilde{s} | s, A)$ in the formulation of Assumption 2, as you have suggested.

---

### Official Review · Reviewer_LEuf · 2022-10-23

**Confidence:** 4
**Correctness:** 3
**Technical Novelty And Significance:** 2
**Empirical Novelty And Significance:** 2
**Recommendation:** 6

**Clarity, Quality, Novelty And Reproducibility:**

This paper is in good shape and the writing is understandable. The authors clearly express their main idea through this article.
This paper extends the previous work on the proof of identifiability to the multimodal setting. To some extent, it can help the research of multimodal representation learning.


**Strength And Weaknesses:**

This paper proves that contrastive learning can block-identify latent factors shared between heterogenous modalities from both theoretical and experimental aspects. The authors describe their settings and experiments in a specific way, which is understandable.
However, I still have some concerns about this paper.
1) Theorem 1 assumes that the number of content variables is known or that it can be estimated. It seems satisfied in the settings of Multimodal3DIdent. But what about more complex natural datasets like COCO? Or some modalities beyond image and text? It needs more in-depth discussions, although the authors have mentioned in the limitation part.
2) In the experimental part, the authors use Convnet for text encoding, which is the same as the convolutional network used for image encoder. Why not use more general and classical sequential neural network, like LSTM or GRU? They are more widely used in multimodal tasks.


**Summary Of The Paper:**

This paper addressed the problem of identifiability for multimodal representation learning and showed that contrastive learning can block-identify latent factors shared between heterogenous modalities. The authors first prove this theorem based on some assumptions. Afterwards, some experimental results are offered to corroborate their theoretical results.

**Summary Of The Review:**

In general, this article is well organized and reasonable, and the conclusion is helpful. But I still have some concerns, see Strength And Weaknesses.

---

> ### Author Response · Authors · 2022-11-19
> **Response to Reviewer LEuf**
>
> > This paper is in good shape and the writing is understandable. The authors clearly express their main idea through this article. This paper extends the previous work on the proof of identifiability to the multimodal setting. To some extent, it can help the research of multimodal representation learning.
>
> Thank you for the positive comments.
>
> In the following, we address your questions and provide new results that demonstrate how the number of content factors can be estimated in practice.
>
> > Theorem 1 assumes that the number of content variables is known or that it can be estimated. It seems satisfied in the settings of Multimodal3DIdent. But what about more complex natural datasets like COCO? Or some modalities beyond image and text? It needs more in-depth discussions, although the authors have mentioned in the limitation part.
>
> Correct, similar to previous work (e.g., Kuegelgen et al., 2021; Lyu et al., 2022), the identifiability result in Theorem 1 assumes that the number of content variables is known or that it can be estimated. We discuss this point in Section 7 and include new results in Appendix B.2 [Figure 7](https://ibb.co/m52thXy), demonstrating that the validation loss can be used to estimate a *range* of values for the number of content variables. Our results suggest that the estimation of the number of content variables is feasible; however, we also point out that more work is required and emphasize that it is an interesting direction for future research.
>
> Similarly, for more complex natural datasets like COCO, one could also measure the validation loss as a function of the encoding size. However, most real-world multimodal datasets (including COCO) provide no *ground-truth* for shared and modality-specific information. Therefore, we cannot use them to draw a clear conclusion about the identifiability of these groups of factors. Instead, our suite of experiments provides full control over the ground-truth content, style, and modality-specific factors, which allows us to evaluate the identifiability precisely and thus provides strong empirical support for our theoretical results.
>
>
> > In the experimental part, the authors use Convnet for text encoding, which is the same as the convolutional network used for image encoder. Why not use more general and classical sequential neural network, like LSTM or GRU? They are more widely used in multimodal tasks.
>
> We decided to use a convolutional neural network for the text modality, since text convolutions provide a simple architecture that works well in practice [1-4] and because our application only requires outputs of fixed length (i.e., not an output for every word in the sequence). In addition, we would like to clarify that we do *not* use the same network for the image and text encoders (see Table 2b).
>
> > In general, this article is well organized and reasonable, and the conclusion is helpful. But I still have some concerns, see Strength And Weaknesses.
>
> Thank you for the positive feedback in the summary of your review. We hope that our reply and new experiments clarify your concerns.
>
>
> $\ $
>
> ---
>
> References:
> 1. Yoon Kim: Convolutional Neural Networks for Sentence Classification. EMNLP 2014.
> 2. Xiang Zhang, Junbo Jake Zhao, Yann LeCun: Character-level Convolutional Networks for Text Classification. NIPS 2015.
> 3. Ye Zhang, Byron C. Wallace: A Sensitivity Analysis of (and Practitioners' Guide to) Convolutional Neural Networks for Sentence Classification. IJCNLP 2017.
> 4. Yoav Goldberg: Neural Network Methods for Natural Language Processing. Morgan & Claypool Publishers 2017.

---

> > ### Comment · Reviewer_LEuf · 2022-11-22
> > **Response to authors**
> >
> > Thanks for the response. After reading all the responses, I decide to keep my score unchanged.

---

> > > ### Author Response · Authors · 2022-11-24
> > > **Thank you for the feedback**
> > >
> > > Thank you for your feedback. Please feel free to let us know if there are other clarifications we can offer.

---

### Official Review · Reviewer_VZnd · 2022-10-24

**Confidence:** 3
**Correctness:** 3
**Technical Novelty And Significance:** 3
**Empirical Novelty And Significance:** 3
**Recommendation:** 6

**Clarity, Quality, Novelty And Reproducibility:**

**Clarity**

The paper is overall clear to the readers. I list some points that are not precise below:

The notation in 5.1 needs clarification. I may miss out on those definitions, but I did not find the definition of $\boldsymbol{a}$ and $B$, which are related to the mean of style distribution.

The presentation of Fig. 3 is a little bit confusing to me. For example, in (a),  it seems the authors mix the results of categorial and numerical attributes in the same plot. Are there any specific reasons to do so? On the one hand, it weakens the legibility of the figure; on the other hand, these metrics are not comparable in the same plot.



**Novelty**

The proposed method has novelty in the context of multi-modal ICA, though there are some existing works in combining self-supervised models with generative models in both single- and multi-modal scenarios.

**Quality**

The proposed method is technically sound. The theoretical and empirical analyses are comprehensive. Some ablation studies may be needed.

**Reproducibility**

The authors include details for the algorithm and model parameterization, which should be useful for reproducing their results. A demo code is also provided.

**Strength And Weaknesses:**

**Strength**

The paper is well-written and organized, making it easy to follow. The proposed generative model is well presented and justified in a mathematical way. Combining the generative model with the contrastive framework is fairly novel in the ICA context. The generative process is reasonable and can be naturally combined with the contrastive framework. The theoretical analysis is well-defined with the given hypothesis. From the empirical perspective, the authors have shown their main experiments reasonably, and the results support the claims of the paper. The details included should be sufficient for reproducibility.

**Weakness**

Although the math analysis is clearly elaborated, the training of the proposed method is not clarified and I need to figure it out by reading their code. Also, after reading through the code, I found some details are omitted in the paper. For example, the generative function f and the contrastive encoder g need to be separately trained (please correct me if I understand it in the wrong way). Maybe adding a paragraph or an algorithm box will better clarify this part.

The practical training loss and the loss used for analysis are not the same. Although these two loss functions have a close correlation, there exists an approximation gap in terms of the number of samples used in the calculation.

The generative process also introduces several variables/hyper-parameters that need to handle. For example, the generative functions $f_1$ and $f_2$ need to be invertible; the dimensionality and covariances of the latent variables need to be carefully chosen. Some additional ablation studies might be needed to address these concerns.

The latent variables, even the whole framework, should be affected by the performance of the generative functions. The authors should provide either theoretical or empirical studies to instruct the impacts. For example, how does the contrastive encoder perform in terms of the reconstruction error of the generative models?

For practical use, since the framework has some assumptions which can hardly be satisfied in real-world cases, it is hard to evaluate the contributions from this perspective. The experiments are conducted at the toy level, and it is unclear what would happen in bigger and more complex data. Some relaxation or approximation are in need in the future.

Some math notations are not fully explained. Please see the clarity part.

**Summary Of The Paper:**

This paper presents a contrastive learning method combined with a generative process where the latent space involves the disentangled latent variables "content", "style" and "domain-specific" elements. The authors provide theoretical analysis using AlignUniform loss, a variant of the InfoNCE loss and demonstrate that contrastive learning in a multi-modality setting could block-identify the common latent attributes across domains like contents. Empirically, the authors conduct simulations with InfoNCE loss on both synthetic and Causal3DIdent data for numerical justifications.

**Summary Of The Review:**

The topic of this paper is interesting in the study of contrastive learning in a multi-modal case. The proposed method is well supported with both theoretical and empirical evidence. Some important perspectives regarding the proposed method are either not clarified or comprehensively studied at this moment.

---

> ### Author Response · Authors · 2022-11-19
> **Response to Reviewer VZnd (part 2/2)**
>
> > The latent variables, even the whole framework, should be affected by the performance of the generative functions. The authors should provide either theoretical or empirical studies to instruct the impacts. For example, how does the contrastive encoder perform in terms of the reconstruction error of the generative models?
>
> Again, we would like to clarify that there is **no generative model** that needs to be trained. This is important and explains why our study does not include ablations for the reconstruction error. We hope that our previous answers already clarify your concerns.
>
> > For practical use, since the framework has some assumptions which can hardly be satisfied in real-world cases, it is hard to evaluate the contributions from this perspective. The experiments are conducted at the toy level, and it is unclear what would happen in bigger and more complex data. Some relaxation or approximation are in need in the future.
>
> Could you please clarify which assumptions cannot be satisfied in real-world cases? In addition to experiments on simulated data, we already provide experiments on a complex multimodal dataset of image/text pairs. We believe that the dataset is representative of real-world use cases and complementary to already existing empirical analyses of models like CLIP (Radford et al., 2021), which are trained on large, curated datasets of image/text pairs. To broaden the scope of our evaluation, we now also provide an additional experiment with pairs of high-dimensional images (Appendix B.2 [Figure 10](https://ibb.co/h9rBZnf)), which shows that block-identifiabilty can also be achieved for complex datasets of image pairs. In terms of realism, our suite of experiments is in line with the state-of-the-art in this field (Kuegelgen et al., 2021; Zimmermann et al., 2021).
>
>
> > The notation in 5.1 needs clarification. I may miss out on those definitions, but I did not find the definition of $a$ and $B$, which are related to the mean of style distribution.
>
> Thank you for pointing this out. For the numerical experiment, the mean of the style distribution is defined as $a + B c$, i.e., a linear transformation of the content variable. For settings with a causal dependence of style on content, we set $a_i, B_{ij} \sim \mathcal{N}(0, 1)$; otherwise, we set $a_i, B_{ij} = 0\ \forall\ i,j$, such that style becomes independent of content. In both cases, the variables are initialized once and remain fixed throughout training. We now include these details in the updated version of our manuscript.
>
>
> > The presentation of Fig. 3 is a little bit confusing to me. For example, in (a), it seems the authors mix the results of categorial and numerical attributes in the same plot. Are there any specific reasons to do so? On the one hand, it weakens the legibility of the figure; on the other hand, these metrics are not comparable in the same plot.
>
> We chose to plot the predictive performance for categorical and numerical variables together in Figure 3a to show the trend jointly across all attributes. We agree that it is not optimal for the comparison of *individual* values across attributes, however, we believe it helps in visualizing the *trend* across all attributes as a function of the encoding size. Further, we explicitly mention the combination of discrete and continuous attributes in the caption of the respective figure, and in Table 3 we specify the distribution of each attribute for the given dataset.
>
>
> > The topic of this paper is interesting in the study of contrastive learning in a multi-modal case. The proposed method is well supported with both theoretical and empirical evidence. Some important perspectives regarding the proposed method are either not clarified or comprehensively studied at this moment.
>
> Thank you for the overall positive feedback in your summary of the review. We hope that our response helps you understand that there is no generative model as part of the learning framework and that this clarifies the missing ablation studies that you originally suggested.

---

> > ### Comment · Reviewer_VZnd · 2022-11-24
> > **Response to the authors**
> >
> > I appreciate the authors' efforts in addressing my concerns. After reading the responses and my co-reviewers' comments, I think a part of my concerns is dissipated except for the following two perspectives:
> >
> > > The gap between theoretical analysis and empirical results.
> >
> > As we agree, the practical training loss and the loss used for analysis are not the same, it is still unclear to me why the theoretical results are invariant to the choice of estimators and how the authors bridge the gap between the theory and the practice. Could you please clarify this part? Moreover, I am a little bit confused about whether this work uses an invertible encoder. It seems your mixing function $f_1$ and $f_2$ are invertible, according to the paper. However, in the authors' response to VsMH, it is claimed a major distinction to Lyu et al., (2022) is the framework studies contrastive learning without the requirement of an invertible $f$.
> >
> > > About the mixing function $f$
> >
> > I understand there is no training for $f_1$ and $f_2$, and I found that in your code, which is why I suggest clarifying this in the paper or adding an algorithm box to help the readers.
> >
> > "How is contrastive learning affected by the invertible functions $f_1$ and $f_2$" is the point that I was trying to make in the review. Based on my understanding, the authors study contrastive learning in a projected space, which is related to the mixing function $f_1$ and $f_2$. From previous studies of contrastive learning, we know the ways of positive pair sampling are different in different data spaces. Thus here I am wondering how the representation quality would vary if we use different parameterizations of $f_1$ and $f_2$.
> >
> > In the provided code, $f_1$ and $f_2$ are instantiated with PCA. Besides PCA, we may wonder if we can use some pre-trained invertible neural networks to parameterize $f_1$ and $f_2$, as PCA might not be very effective on more complex real-world data such as ImageNet1K. In this case, the dimensionality of the projected space and how well the network approximates the invertible function need more discussion.

---

> > > ### Author Response · Authors · 2022-11-26
> > > **Thank you for your questions**
> > >
> > > > After reading the responses and my co-reviewers' comments, I think a part of my concerns is dissipated except for the following two perspectives
> > >
> > > Thank you for your feedback. We are delighted to see that our reply resolved a large part of your concerns. In the following, we seek to address all remaining questions.
> > >
> > > > it is still unclear to me why the theoretical results are invariant to the choice of estimators and how the authors bridge the gap between the theory and the practice. Could you please clarify this part?
> > >
> > > Thank you for the question. We extend our answer from our previous reply: We bridge the gap between theory and practice by providing multiple experiments that demonstrate how the theoretical guarantees (for the objective with $K \to \infty$) translate to practical settings where we use **suitable estimators** with finite $K$. In particular, we **empirically verify** that the ground-truth content variables are indeed block-identified using numerical simulations, and we corroborate our findings on complex multimodal datasets of image/text and image/image pairs. In Section 2.2, we already discuss that **the estimator asymptotically approximates the theoretical objective** for $K \to \infty$. To understand why finite-sample guarantees are non-trivial for our problem setup, please refer to [our answer to Reviewer VsMH](https://openreview.net/forum?id=U_2kuqoTcB&noteId=6BWMLsyg-hV).
> > >
> > >
> > > > I am a little bit confused about whether this work uses an invertible encoder. It seems your mixing function $f_1$ and $f_2$ are invertible, according to the paper.
> > >
> > > To be clear, **we do not use invertible encoders**, as noted in previous replies ([1](https://openreview.net/forum?id=U_2kuqoTcB&noteId=dBb5rJ_peS5), [2](https://openreview.net/forum?id=U_2kuqoTcB&noteId=6BWMLsyg-hV)) and specifically in our [new reply to Reviewer VsMH](https://openreview.net/forum?id=U_2kuqoTcB&noteId=6BWMLsyg-hV). Generally, the *mixing functions* (i.e., not the encoders) have to be invertible, since otherwise the recovery of the ground-truth latent variables would be theoretically impossible, as there would be an unavoidable information loss. Invertibility of the *mixing functions* is a **standard assumption** in the nonlinear ICA literature (e.g., Hyvärinen et al., 2019; Gresele et al., 2019; Khemakhem et al., 2020).
> > >
> > >
> > > > I understand there is no training for $f_1$ and $f_2$, and I found that in your code, which is why I suggest clarifying this in the paper or adding an algorithm box to help the readers.
> > >
> > > Since it was previously stated that "*the generative function $f$ and the contrastive encoder $g$ need to be separately trained (please correct me if I understand it in the wrong way)*", we assumed that this point was unclear. We apologize for any potential misunderstanding and will ensure to clarify this in our paper.
> > >
> > >
> > > > "How is contrastive learning affected by the invertible functions $f_1$ and $f_2$" is the point that I was trying to make in the review. [...] Thus here I am wondering how the representation quality would vary if we use different parameterizations of $f_1$ and $f_2$. In the provided code, $f_1$ and $f_2$ are instantiated with PCA. Besides PCA, we may wonder if we can use some pre-trained invertible neural networks to parameterize $f_1$ and $f_2$, as PCA might not be very effective on more complex real-world data such as ImageNet1K.
> > >
> > > We would like to clarify that, generally, the mixing functions are **not** a design consideration in the implementation. Generally, the encoders take observations (e.g., high-dimensional image or text data) as input, hence **we do not study contrastive learning in a projected space**. Only for the numerical simulation, we artificially construct the mixing functions using *nonlinear, invertible MLPs* (i.e., not PCA, which would be a linear transformation), similar to previous work (Zimmermann et al., 2021; von Kügelgen et al., 2021). We will publish the dataset and code for the image/text experiment after the reviewing phase.
> > >
> > >
> > > > In this case, the dimensionality of the projected space and how well the network approximates the invertible function need more discussion.
> > >
> > > As already noted in the answer above, we do not study contrastive learning in a projected space, which explains why we do not use different parameterizations of $f_1$ and $f_2$. Since **the encoders take observations as input**, this should also explain why we do not discuss the dimensionality of the "projected space", which is simply the dimensionality of the observation space (e.g., the number of pixels in an image).
> > >
> > > We would like to thank the reviewer for all questions. We hope that our reply resolves all remaining questions and, in that case, would greatly appreciate an adjustment of the score. Please do not hesitate to let us know if there are any further clarifications we can offer.

---

> > > > ### Comment · Reviewer_VZnd · 2022-12-03
> > > > **Response to the authors**
> > > >
> > > > Thanks for the responses and clarifications. I updated the score after reading the revision with the latest responses as it is clear to me now. I appreciate it the authors can incorporate the reviewers' suggestions in the final version.

---

> > > > > ### Author Response · Authors · 2022-12-05
> > > > > **Thank you for your response**
> > > > >
> > > > > Thank you for your response. We are delighted to hear that our replies have resolved your concerns and greatly appreciate that you have increased the score. We will make sure to incorporate the reviewers' suggestions in the final version.

---

> ### Author Response · Authors · 2022-11-19
> **Response to Reviewer VZnd (part 1/2)**
>
> > The theoretical analysis is well-defined with the given hypothesis. From the empirical perspective, the authors have shown their main experiments reasonably, and the results support the claims of the paper. The details included should be sufficient for reproducibility.
>
> Thank you for the positive feedback.
>
> In the following, we address your concerns and clarify one likely misunderstanding that comes up multiple times in your review. We would like to clarify that the model is trained only using a contrastive loss; it is **not a generative model** and does not reconstruct the observations. This is important, because it explains why our study does not include certain ablations (e.g., reconstruction error).
>
>
> > Combining the generative model with the contrastive framework is fairly novel in the ICA context. The generative process is reasonable and can be naturally combined with the contrastive framework.  [...] The proposed method has novelty in the context of multi-modal ICA, though there are some existing works in combining self-supervised models with generative models in both single- and multi-modal scenarios.
>
> While we appreciate your positive comments regarding the novelty of our work, we would like to clarify that the combination of a generative model with contrastive learning is not the topic of our paper. Instead, we show that contrastive learning can block-identify latent factors shared between heterogeneous modalities (e.g., images and captions), even in the presence of nontrivial statistical and causal dependencies.
>
>
> > Although the math analysis is clearly elaborated, the training of the proposed method is not clarified [...]. For example, the generative function f and the contrastive encoder g need to be separately trained (please correct me if I understand it in the wrong way).
>
> We would like to clarify that the model is trained only using a contrastive loss. Specifically, we only train the encoder via optimization of the contrastive objective in Equation (4), which is also what is implemented in the supplemented code. For example, in the numerical simulation, we initialize the *ground-truth* data-generating process as a pair of invertible mixing functions $f_1$ and $f_2$. After initialization, these mixing functions remain fixed (see line 106 in the file `invertible_network_utils.py`); only the encoders $g_1$ and $g_2$ are trained. Maybe the source of the misunderstanding is that we unfortunately swap the notation compared to the paper (in the code, `g_i` denotes $f_i$, whereas `f_i` denotes $g_i$); in this case, we apologize for the inconsistent notation. We hope that this clarifies your questions and uncertainties regarding the training of the model.
>
>
> > The practical training loss and the loss used for analysis are not the same. Although these two loss functions have a close correlation, there exists an approximation gap in terms of the number of samples used in the calculation.
>
> Correct, in the theoretical analysis we use the asymptotic form of the objective (where the number of negative samples $K$ goes to infinity), while we use a finite-sample estimator (i.e., InfoNCE with finite $K$) for the experiments. However, the experiments validate our theoretical results *despite* the use of estimators.  Hence, we bridge the gap between theory and practice by providing multiple experiments that demonstrate how the theoretical guarantees translate to practical settings (with finite $K$).
>
> > The generative process also introduces several variables/hyper-parameters that need to handle. For example, the generative functions and need to be invertible; the dimensionality and covariances of the latent variables need to be carefully chosen. Some additional ablation studies might be needed to address these concerns.
>
> Please note that the dimensionality and covariances of the latent variables do *not* need to be carefully chosen, as they do not represent modeling decisions but assumptions on the ground-truth data-generating process. Without any assumptions, the latent variables are known to be non-identifiable for a nonlinear data generating process (Hyvärinen and Pajunen, 1999). We clearly state the assumptions that are necessary for identifiability in the multimodal setting and empirically verify the identifiability results under several variants of these assumptions (e.g., statistical and causal dependencies) and even violations thereof (e.g., discrete factors). On the other hand, there are also model hyper-parameters, such as the size of the encoding, for which we already provide an ablation in Figure 3.

---

### Official Review · Reviewer_akaD · 2022-10-25

**Confidence:** 4
**Correctness:** 3
**Technical Novelty And Significance:** 3
**Empirical Novelty And Significance:** 3
**Recommendation:** 6

**Clarity, Quality, Novelty And Reproducibility:**

The paper is largely clear with clear figures and exposition. It was a joy reading this paper. The paper makes important contributions to the understanding of contrastive, multiview and multimodal learning, with important insights for future work backed up by theory and experiments.

**Strength And Weaknesses:**

Strengths:
1. The paper is well motivated and described. The ideas are clear and experiments are on several large multimodal datasets.
2. The paper is largely clear with clear figures and exposition. It was a joy reading this paper.
3. The paper makes important contributions to the understanding of contrastive, multiview and multimodal learning, with important insights for future work.

Weaknesses:
1. The synthetic and real data experiments should vary the contribution across modality-general style and content and modality-specific to better determine how one can identify the important modality-specific factors across different settings. For example does identifiability get harder with more modality-general information? or more modality-specific information?
2. More real-world multimodal datasets can be experimented with, see https://arxiv.org/abs/2107.07502, especially datasets with a broad range of ratios in modality-general vs modality-specific information.
3. The theory, while seems to be right and justified, is rather high-level, and does not delve into the details of learning and optimization (e.g., guarantees on how to learn such disentangled latent variables, or to encode different modalities). A lot of the proof follows by extending prior work in multiview ICA combined with the newly made assumptions in the paper, which is okay, but seems rather high-level. What if the assumptions are not made? What are some directions to further improve this line of theoretical study?

**Summary Of The Paper:**

This paper studies contrastive learning (e.g., in representation learning with multiview data or multimodal image/caption pairs). This paper presents new identifiability results for multimodal contrastive learning by extending recent work in identifiability results for multiview contrastive learning. Their main contribution is to extend the analysis of multiview non-linear ICA with a single shared latent variable to a multimodal case where it is possible to have a disentangled set of latent variables across modality-shared and modality-specific parts, which all jointly generate the observed multimodal data. Their main result shows that it is still possible to recover shared factors in this new setup, specifically that one can block-identify latent factors shared between modalities. They empirically verify these results on synthetic data and multimodal image/text paired data.

**Summary Of The Review:**

The paper makes important contributions to the understanding of contrastive, multiview and multimodal learning, with important insights for future work backed up by theory and experiments, but there can be deeper empirical experiments and concluded insights to be more impactful.

---

> ### Author Response · Authors · 2022-11-19
> **Response to Reviewer akaD**
>
> > The paper is largely clear with clear figures and exposition. It was a joy reading this paper. The paper makes important contributions to the understanding of contrastive, multiview and multimodal learning, with important insights for future work backed up by theory and experiments.
>
> Thank you for your positive comments regarding the relevance and clarity of our work.
>
> In the following, we address your questions and focus on your main point of criticism, i.e., that *“there can be deeper empirical experiments and concluded insights to be more impactful”*, which we seek to address in our answer and by providing new experiments.
>
> > The synthetic and real data experiments should vary the contribution across modality-general style and content and modality-specific to better determine how one can identify the important modality-specific factors across different settings. For example does identifiability get harder with more modality-general information? or more modality-specific information?
>
> Thank you for suggesting new experiments to test the contribution across the different groups of factors. We include a new ablation study (Appendix B.2 [Figure 6](https://ibb.co/WcFFtfK)), where we vary the contribution of style and modality-specific information relative to content. We are pleased to report that the new results corroborate that block-identifiability of content still holds when we vary the relative contribution across these groups of factors. Our results even provide some intuition about the sample complexity: the identification might indeed get harder with more style or modality-specific information, as the models require more iterations to converge in these settings (Subfigures 6b and 6d). Yet, at the end of training, block-identifiability of content is still satisfied. We point out the possible effect on the sample complexity in Appendix B.2 where we discuss the results from Figure 6, and we reference these new results in the discussion (Section 7).
>
> > More real-world multimodal datasets can be experimented with, see https://arxiv.org/abs/2107.07502, especially datasets with a broad range of ratios in modality-general vs modality-specific information.
>
> We greatly appreciate your suggestion to include new datasets and thank you for pointing out the MultiScale benchmark in particular. While we think it would be useful to experiment with additional real-world data, the vast majority of real-world multimodal datasets (including the MultiScale benchmark) provide **no absolute ground-truth** for shared and modality-specific information, but only human-generated labels. Instead, our suite of experiments provides full control over the ground-truth content, style, and modality-specific information, which allows us to evaluate the identifiability precisely and thus provides strong empirical support for our theoretical results.
>
> > The theory, while seems to be right and justified, is rather high-level, and does not delve into the details of learning and optimization (e.g., guarantees on how to learn such disentangled latent variables, or to encode different modalities).
>
> Thank you for acknowledging our theoretical results. While our theory does not yield finite-sample guarantees for the optimization (as you correctly point out), we actually provide guarantees for the asymptotic case (i.e., when the number of negative samples $K$ goes to infinity) to achieve a certain form of disentanglement (i.e., block-identifiability). Moreover, we bridge the gap between theory and practice by providing multiple experiments that demonstrate how the theoretical guarantees translate to practical settings (with finite $K$) for which the ground-truth content variables are indeed block-identified.
>
> > A lot of the proof follows by extending prior work in multiview ICA combined with the newly made assumptions in the paper, which is okay, but seems rather high-level. What if the assumptions are not made? What are some directions to further improve this line of theoretical study?
>
> In Section 7, we already discuss the key assumptions, possible violations, and directions for future work. To address violations, we include dimensionality ablations where we vary the encoding size (Figure 3) as well as experiments with discrete latent factors (Section 5.2; Appendix B.2). Moreover, we would like to clarify that our work *relaxes* the assumptions from previous work and we point out concrete research directions (e.g., further relaxations) in Section 7.
>
> > The paper makes important contributions to the understanding of contrastive, multiview and multimodal learning, with important insights for future work backed up by theory and experiments, but there can be deeper empirical experiments and concluded insights to be more impactful.
>
> Thank you for the positive feedback in the summary of your review. We hope that our reply and our additional experiments clarify your concerns regarding the impact of the empirical insights.

---

> > ### Comment · Reviewer_akaD · 2022-11-30
> > **thanks for your response**
> >
> > Thank you authors for the additional experiments and revisions to the paper, I am happy to increase my score to 6.

---

> > > ### Author Response · Authors · 2022-12-01
> > > **Thank you for the feedback**
> > >
> > > Thank you for your feedback and for raising the score. We are delighted to see that the additional experiments and revisions resolved your concerns. Please feel free to let us know if there are any further clarifications we can offer.

---

### Official Review · Reviewer_VsMH · 2022-11-13

**Confidence:** 5
**Correctness:** 3
**Technical Novelty And Significance:** 2
**Empirical Novelty And Significance:** Not applicable
**Recommendation:** 6

**Clarity, Quality, Novelty And Reproducibility:**

Clarity: The paper is in general written in a clear and easy-to-follow manner. However, the technical assumptions may benefit from more discussion and explanation.

Quality: The paper has a good quality in terms of technical contents and presentation.

Novelty:  One of the major concern of the reviewer is that the paper has missed a very relevant existing work (Lyu et al 2022) that has established block-identifiability of a similar model. But (Lyu et al 2022) was not discussed or mentioned in this paper. This makes the contributions of this work relative to (Lyu et al 2022) a bit unclear.

Reproducibility: no major issue identified.

**Strength And Weaknesses:**

Strength

- The considered problem is interesting, and meaningful. Compared to the work of (von Kugelgen et al. 2021), the two different generative functions for two views, respectively, can be applied to a wider range of applications, e.g., text v.s. image type multiview data.

- The paper is well-written.

Weakness

- Some technical assumptions are not easy to follow. Although the paper is in general pleasant to read, the part around Assumptions 1-2 are a bit hard to digest. Some more explanations and relating to physical meaning may help.

- There work only established block identifiability of the shared components, but did not consider identifying the private components.

- The work's identifiability result was established by assuming that infinite data samples are available, which is not realistic.

**A major concern is that discussion regarding a very relevant existing work is missing**. The submission claims that a major contribution is that the work established block-identifiability of a multiview nonlinear mixture model when different modalities have different generating functions. However, the reviewer hopes to draw attention of the authors to the following paper:

(Lyu et al 2022) Lyu, Qi, Xiao Fu, Weiran Wang, and Songtao Lu. "Understanding latent correlation-based multiview learning and self-supervision: An identifiability perspective." In ICLR 2022

This recent work (Lyu et al 2022) has already established a similar identifiability result under a similar multiview model with different view-generating mechanisms. The work was not discussed in this submission, but it is very relevant:

Some similarities are listed as follows:

- Using this work’s notation, the generative model with x_1 = f_1(c,m_1) and x_2=f_2(c,m_2) was considered in (Lyu et al 2022), and the identifiability of c and m_1 and m_2 were established using a nonlinear multimodal learning criterion. This is quite similar to the generative model considered in this submission, where the latents can be split to shared and view-specific components.

- The identifiability in both works were established via latent shard component matching with invertibility regularization. The ideas share the same spirit.

- The criterion in (Lyu et al 2022) was also related to a multimodal contrastive learning criterion in Appendix I.2. This makes the learning criteria of the two works even more similar to each other.

I hope to mention that some additional contributions in (Lyu et al 2022):

- Unlike this paper that only considers infinite-sample cases, (Lyu et al 2022) also has finite-sample identifiability analysis.

- Unlike this submission that only considers block-identifying of the shared components, (Lyu et al 2022) also proved that the private components can be block-identified under some conditions.

The reviewer also acknowledges that there are some differences:

- The model considered in this submission is slightly different, where the private components are split into s and m_i for i=1,2. But the model in (Lyu et al 2022) does not have the s part.
- The assumptions used to establish identifiability are different in the two works.
- The proof techniques are not the same.
- Due to the introduction of s in this submission, the framework in this submission is perhaps more suitable for causal learning.

Given the similarity in terms of the generative models and the learning criteria in both works, the fact that (Lyu et al 2022) is completely missing from the bibliography of this submission is unfortunate. Proper discussions regarding the very relevant prior work (Lyu et al 2022) and existing block-identifiability results may make the contributions of this submission clearer.


**Summary Of The Paper:**

The work extends the contrastive learning based nonlinear mixture identification framework in (von Kugelgen et al. 2021) and considers a case where the two modalities have two different generative functions. The major contribution is the models’ identifiability analysis. The setting and the analysis can be considered as extensions of those in (von Kugelgen et al. 2021). A multimodal nonlinear mixture model is adopted, and a contrastive learning-based criterion is analyzed. The major claimed contribution lies in analyzing the case where two views have different generative nonlinear functions, but in (von Kugelgen et al 2021) assumed that the two views share the same generative nonlinear system.

**Summary Of The Review:**

This is a well-written and well-organized paper. The major concern lies in its novelty relative to an existing work. This should be properly acknowledged and discussed, which would have made the contributions of this work clearer.

---

> ### Author Response · Authors · 2022-11-19
> **Response to Reviewer VsMH (part 2/2)**
>
> In the following, we address individual points from your review in more detail.
>
> > * Some technical assumptions are not easy to follow. Although the paper is in general pleasant to read, the part around Assumptions 1-2 are a bit hard to digest. Some more explanations and relating to physical meaning may help.
> > * The paper is in general written in a clear and easy-to-follow manner. However, the technical assumptions may benefit from more discussion and explanation.
>
> Thank you for the feedback. We already try to convey the intuition behind Assumptions 1 and 2 in the surrounding text, where we explain style changes and conditions (i)--(iii) from Assumption 2 in more detail. Could you please specify which statements exactly are not easy to follow such that we can improve the text?
>
> Additionally, we provide some intuition about the different groups of latent variables in the last paragraph of Section 3.1, where we explain that the content invariance describes a shared phenomenon that is not directly observed but manifests in the observations from both modalities, that style changes describe shared influences that are not robust across modalities (e.g., non-invertible transformations such as data augmentations, or non-deterministic effects of an unobserved confounder), and modality-specific factors can be viewed as variables that describe the inherent heterogeneity of each modality (e.g., background noise).
>
> > There work only established block identifiability of the shared components, but did not consider identifying the private components.
>
> This is true, but it is also not the goal of our paper. We study contrastive learning in the multimodal setup with nontrivial dependencies between latent factors. Specifically, we show that contrastive learning can block-identify the invariant components (i.e., content) and exclude both private (i.e., modality-specific) and stochastically shared components (i.e., style) from heterogeneous modalities even when there are nontrivial statistical and causal dependencies between latents.
>
> > The work's identifiability result was established by assuming that infinite data samples are available, which is not realistic.
>
> We agree that finite-sample guarantees are desirable and would be interesting to investigate for future work; however, such results are non-standard and highly nontrivial under our assumptions, i.e., assuming causal dependencies, style-perturbations of arbitrary strength, and non-invertible encoders. Moreover, we empirically verify our theoretical results with numerical simulations and corroborate our findings on a complex multimodal dataset of image/text pairs.
>
> > The criterion in (Lyu et al 2022) was also related to a multimodal contrastive learning criterion in Appendix I.2. This makes the learning criteria of the two works even more similar to each other.
>
> In Appendix I.2, Lyu et al. (2022) propose a slack variable based design that is applicable to multimodal contrastive learning. Even though their learning criterion was not empirically validated, we believe that it shows an elegant design that could lead to interesting applications for multimodal contrastive learning with more than two modalities, which we point out for future work in Section 7. However, we study a different learning criterion (i.e., the symmetrized InfoNCE loss), which is also more closely related to the objectives used in existing applications of multimodal contrastive learning (e.g., Zhang et al., 2020; Radford et al., 2021). Using this objective, we provide an extensive empirical evaluation with numerical simulations and applications to complex multimodal data.
>
> > The identifiability in both works were established via latent shard component matching with invertibility regularization. The ideas share the same spirit.
>
> We would like to clarify that our block-identifiability result (Theorem 1) applies to non-invertible encoders with entropy regularization, which is distinct from Theorem 1 in Lyu et al. (2022), which explicitly assumes invertible encoders. In Appendix I.1, Lyu et al. (2022) discuss how invertibility can be achieved via entropy regularization, but their result does not cover the cases of stochastically shared components (i.e., style) or causal dependencies from content to style. Moreover, the distinction between the two setups---invertible encoders vs. non-invertible encoders with entropy regularization---was already discussed in previous work (von Kügelgen et al., 2021).
>
> > This is a well-written and well-organized paper. The major concern lies in its novelty relative to an existing work. This should be properly acknowledged and discussed, which would have made the contributions of this work clearer.
>
> Thank you for your positive feedback on the quality and clarity of our manuscript. We hope that our reply clarifies your concerns about the contribution of our work relative to the work of Lyu et al. (2022), which is now integrated into the updated version of our manuscript.

---

> > ### Comment · Reviewer_VsMH · 2022-11-22
> > **... continuing my reply**
> >
> > *Technical assumptions*: Other than the introduction of m_1 and m_2, the model and assumptions in this work are largely similar to those in (von Kügelgen et al., 2021). It may benefit the clarity of the work how the existence of m_1 and m_2 help with establishing the block-identifiability of c under the setting of this work. In other words, can the proof still hold if m_1 and m_2 do not exist?
> >
> >
> > *Technical assumptions*: It may be helpful to clarify why using a specific type of graphic model, i.e.,  c---> s---->\tilde{s}. For example, one could argue s <--- c ---> \tilde{s} would have also made sense. Under the current setting, it is unclear how to designate roles to each view (or does it matter in practice?).
> >
> >
> > *Non-invertible encoder*: It may be a bit misleading to claim that this work does not need invertible encoder, as the entropy regularization is meant to ensure the encoder to be invertible. The invertibility is naturally needed by the assumption that the generative functions are invertible nonlinear systems. It was an elegant proof from (von Kügelgen et al., 2021) showing that the constrastive loss implicitly ensures invertibility. But still, the invertibility is not circumvented in implementation.
> >
> > Again, I think this work is an interesting extension of existing works. Improving presentation accuracy/clarity and having more in-depth discussion on the technical assumptions may make it stronger.

---

> > > ### Author Response · Authors · 2022-11-24
> > > **Clarification of technical assumptions**
> > >
> > > Thank you for engaging in the discussion. In the following, we seek to address all of your questions regarding the technical assumptions.
> > >
> > > > Other than the introduction of m_1 and m_2, the model and assumptions in this work are largely similar to those in (von Kügelgen et al., 2021).
> > >
> > > Apart from the introduction of $m_1$ and $m_2$, our work also considers modality-specific mixing functions and distinct encoders, as we already discuss in our paper.
> > >
> > > > It may benefit the clarity of the work how the existence of m_1 and m_2 help with establishing the block-identifiability of c under the setting of this work. In other words, can the proof still hold if m_1 and m_2 do not exist?
> > >
> > > Thank you for the question. In short, **our proof still holds** if $m_1$ and $m_2$ do not exist. For further details, please refer to [our reply to Reviewer YHeS](https://openreview.net/forum?id=U_2kuqoTcB&noteId=vMy7DaiFaXB), where we already addressed this question.
> > >
> > >
> > > > Technical assumptions: It may be helpful to clarify why using a specific type of graphic model, i.e., c---> s---->\tilde{s}. [...] Under the current setting, it is unclear how to designate roles to each view (or does it matter in practice?).
> > >
> > > Thank you for your question. Indeed, it **does not matter** how to designate roles to each view or modality. In Appendix A.2, we already explain how our model from Figure 1 can be adapted to describe a *symmetric* generative process $z_1 \gets z \to z_2$.
> > >
> > > > For example, one could argue s <--- c ---> \tilde{s} would have also made sense.
> > >
> > > Would you mind to explain in which settings this relation would provide an additional benefit? The relation would clearly require different assumptions to ensure content invariance. We would leave it as an opportunity for future work.
> > >
> > >
> > > > The invertibility is naturally needed by the assumption that the generative functions are invertible nonlinear systems
> > >
> > > The assumption of an invertible generative process is required to ensure that the observations retain all information from the latent variables, i.e., that there is no information loss. It is a standard assumption in nonlinear ICA literature and, importantly, it is independent of any assumptions on the *encoders*.
> > >
> > >
> > > > Non-invertible encoder: It may be a bit misleading to claim that this work does not need invertible encoder, as the entropy regularization is meant to ensure the encoder to be invertible. The invertibility is naturally needed by the assumption that the generative functions are invertible nonlinear systems. It was an elegant proof from (von Kügelgen et al., 2021) showing that the constrastive loss implicitly ensures invertibility. But still, the invertibility is not circumvented in implementation.
> > >
> > > We respectfully disagree on this point and do not think that "misleading" is the right word in this context, because our assumptions in Theorem 1 clearly do *not* require an invertible encoder *at initialization*. From a practical perspective, this distinction is important because it **justifies using an arbitrary, non-invertible encoder in the implementation**, and we use non-invertible encoders throughout our experiments. In the optimum, it is true that entropy regularization implicitly ensures that the encoder is invertible, which is a **consequence** of block-identification, not a precondition. On the other hand, if one would *assume* an invertible encoder (as in Theorem 1 from Lyu et al., 2022), this can restrict the choice of encoder architectures (e.g., normalizing flows) or require additional loss terms (e.g., reconstructions) which incentivize but do not neccesarily guarantee invertibility. Similar arguments exist in previous work, e.g., see page 4 in Gresele et al. (2019) or page 7 in von Kügelgen et al. (2021). Hence, we agree with previous work and think that---both from a theoretical and practical standpoint---it is **important to distinguish** between using entropy regularization, which implicitly ensures invertibility in the optimum, versus *assuming* invertibility from the onset.
> > >
> > >
> > > > Again, I think this work is an interesting extension of existing works. Improving presentation accuracy/clarity and having more in-depth discussion on the technical assumptions may make it stronger.
> > >
> > > Thank you for acknowledging the relevance of our work and for engaging in this discussion. We find it very helpful to understand how to enhance the presentation of the assumptions. Please feel free to let us know if there are other clarifications we can offer.

---

> > > > ### Comment · Reviewer_VsMH · 2022-11-29
> > > > **some more comments**
> > > >
> > > > Re: m_1 and m_2:  I agree that the existence of m_i does not quite affect the proof. The proof that h_i(x) will not include things from m_i uses block independence. This part is similar to what Lyu et al 2022 used.
> > > >
> > > > Re: "benefits of s <--- c ---> \tilde{s}": My sense is that this is perhaps a possibility, which says given the contents, the styles in each views are conditionally independent. This shows dependence between style and content, but says the styles of different modalities do not necessarily need to affect each other, given the content.
> > > >
> > > >  Re: ``misleading'': I agree with the authors: ``misleading'' should not be the correct word. The fact that the implementation does not need an invertible encoder is not a misrepresentation.
> > > >
> > > > I have re-read the work again and have some more comments:
> > > >
> > > > 1) The proof technique largely follows von Kügelgen et al. (2021), and thus the contribution in terms of analytical methodology seems to be limited. However, the new problem setting is more widely applicable. The observation that the proof in von Kügelgen et al. (2021) can be extended to the settings in this case is not unuseful. Hence, I would lift my score to 6.
> > > >
> > > > 2) This is perhaps minor and can be fixed more easily: Some comments and categorization regarding prior works may not be exactly accurate. For example, von Kügelgen et al. (2021), Lyu and Fu (2020) and Lyu et al 2022 were put under "Multiview nonlinear ICA". But all these works did not assume that the latents were mutually independent.  Locatello et al., (2020) was also put under multiview ICA, but this paper's setting may also cover other cases, e.g., slow-changing settings in a single view case, as long as there is weak supervision.

---

> > > > > ### Author Response · Authors · 2022-12-01
> > > > > **Thank you for the constructive discussion**
> > > > >
> > > > > We thank the reviewer for the valuable feedback and for raising the score. We are delighted to see that the concerns regarding the contribution of our work were resolved, and we greatly appreciate that the reviewer recognizes the utility and wide applicability of our problem setting.
> > > > >
> > > > > > Re: "benefits of s <-- c -->\ts":
> > > > >
> > > > > Thank you for clarifying your original suggestion. We agree that such a setting can be an interesting direction for future work to further relax the assumptions.
> > > > >
> > > > > > Re: minor comments
> > > > >
> > > > > We greatly appreciate these points and will integrate them into the manuscript.
> > > > >
> > > > > Again, we would like to thank the reviewer for the valuable feedback and for the constructive discussion.

---

> ### Author Response · Authors · 2022-11-19
> **Response to Reviewer VsMH (part 1/2)**
>
> > This is a well-written and well-organized paper. The major concern lies in its novelty relative to an existing work. This should be properly acknowledged and discussed, which would have made the contributions of this work clearer.
>
> Thank you for highlighting the interesting and highly relevant work of Lyu et al. (2022) and for the detailed summary of their contributions. We clarify our contribution in the rebuttal and have already integrated the study into the updated version of our manuscript. We appreciate your positive feedback regarding the clarity and quality of our manuscript and that you also point out some of the differences between our paper and the work from Lyu et al. (2022) in your review.
>
> $\ $
>
> ### Distinction from Lyu et al. (2022)
>
> In the following, we clarify the contribution of our work relative to the work of Lyu et al. (2022). Although some of our assumptions are similar, our work tackles a different problem: we deliberately focus on block-identifiability of content in the multimodal setup with statistical and causal dependencies between latent variables. In contrast, the work of Lyu et al. (2022) applies to a more restricted setting with mutually independent groups of variables, which is more closely related to the classical setup of multi-view nonlinear ICA (Gresele et al. 2019; Lyu and Fu, 2020), which we already discuss in our manuscript. In summary, we draw **two important distinctions**: First, our identifiability result holds under a more general set of assumption than the respective result from Lyu et al. (2022). Second, our model is conceptually different from the model in Lyu et al. (2022), even if we consider the most similar special case of their model.
>
> **Major distinction 1:** Our identifiability result from Theorem 1 is more general than Theorem 1 from Lyu et al. (2022), because our result rests on a weaker set of relevant assumptions:
> - Our model allows for style factors $s, \tilde{s}$, which enables the modeling of non-deterministically shared information, such as perturbations $s \to \tilde{s}$ or non-deterministic confounding of the form $s \gets u \to \tilde{s}$. These cases are not covered by the assumptions in Lyu et al. (2022).
> - Our model allows for causal dependencies $c \to s$, which can trivially be extended to causal dependencies $c \to m_1$. In contrast, Lyu et al. (2022) assume mutually independent groups of latent variables, which is a special case of our modeling assumptions. The absence of causal dependencies makes their work more similar to the classical setup of multi-view nonlinear ICA (Gresele et al., 2019; Lyu and Fu, 2020), which we already discuss in the related work section.
> - The above relaxations are relevant for multimodal learning because many applications exist where latent factors are stochastically shared between modalities but are not strictly invariant or modality-specific. This setting is *not* covered by the assumptions from Lyu et al. (2022). In contrast, our work shows that stochastically shared information is discarded so that block-identifiability of invariant factors can be achieved in a more general setup with statistical and causal dependencies between modalities.
>
> **Major distinction 2:** Our model is conceptually different from the model in Lyu et al. (2022), even if we consider the most similar special case of their model:
> - We focus on contrastive learning with non-invertible encoders, whereas the work of Lyu et al. (2022) studies latent correlation maximization assuming invertible encoders. The supplementary results from Appendix I.1 in Lyu et al. (2022) do not consider the case of stochastically shared components (i.e., style) as described under major distinction 1. The conceptual distinction between the two setups---invertible encoders vs. non-invertible encoders with additional entropy regularization---was already discussed in previous work (von Kügelgen et al., 2021).
> - Crucially, the conceptual distinction applies even if we consider only the latent correlation maximization objective from Lyu et al. (2022), since Theorem 1 from Lyu et al. (2022) assumes invertible encoders and, additionally, that the number of view-specific components $D_q$ is known. For the full objective from Lyu et al. (2022), which they use to disentangle shared and view-specific information, there are further distinctions to be drawn, as their objective introduces additional loss terms for reconstructions and independence-enforcing constraints that require a nontrivial min-max optimization.
> - The learning criterion introduced in Appendix I.2 from Lyu et al. (2022) relates multimodal contrastive learning to latent correlation maximization, but it leads to a slack variable based design that is not empirically validated. In contrast, we study a symmetrized learning criterion, which is also more closely related to the objectives used in existing applications of multimodal contrastive learning (e.g., Zhang et al., 2020; Radford et al., 2021).

---

> > ### Comment · Reviewer_VsMH · 2022-11-22
> > **Thanks for the detailed reply**
> >
> > I'd like to thank the authors for the detailed reply. Here, I follow up with the following discussion:
> >
> > *Major distinction 1*: I agree that the model in this work considers causal dependence, which is an interesting perspective. Thanks for the clarification.
> >
> > *Major distinction 2*: I think Lyu et al's method does not need the additional loss (what they call R) and min-max optimization if they only need to extract the shared components. If only the shared part is needed, Lyu et. al.'s method only has $\|| f_1(x_1) - f_2(x_2)\||^2$ + invertibility of $f_1\& f_2$, where the invertibility can be promoted by the entropy regularization as used in this work. This part is particularly similar to the main formulation of this work.  I suggest the authors to have a little more comments on this connection. Showing more connections will strengthen this work and reveal more insights.

---

> > > ### Author Response · Authors · 2022-11-24
> > > **Thank you for the discussion**
> > >
> > > Thank you for engaging with us in the discussion. We greatly appreciate that you acknowledge the first major distinction between our work and the work of Lyu et al. (2022).
> > >
> > > > I think Lyu et al's method does not need the additional loss (what they call R) and min-max optimization if they only need to extract the shared components.
> > >
> > > Thank you for your comment. We agree and **already acknowledged this aspect** in our previous reply (point 2 under [major distinction 2](https://openreview.net/forum?id=U_2kuqoTcB&noteId=dBb5rJ_peS5)). We further elaborate on the distinction between using entropy regularization versus assuming invertible encoders in [our reply](https://openreview.net/forum?id=U_2kuqoTcB&noteId=YEr042dSryI) to another comment of yours.
> > >
> > > > If only the shared part is needed, Lyu et. al.'s method only has $\vert \vert f_1(x) - f_2(x) \vert \vert^2$ + invertibility of $f_1$ & $f_2$, where the invertibility can be promoted by the entropy regularization as used in this work. This part is particularly similar to the main formulation of this work. I suggest the authors to have a little more comments on this connection.
> > >
> > > Thank you for pointing out which connection might not be sufficiently clear. To clarify, in Equation (5), we define the asymptotic objective
> > > $$ E \left[ \vert \vert g_1(x_1) - g_2(x_2) \vert \vert_2 \right] + \frac{1}{2} \big( H(g_1(x_1)) + H(g_2(x_2)) \big)$$
> > > which is similar to the objective in Equation (I.2a) from Lyu et al. (2022), which they introduce in Appendix I.1. In particular, Lyu et al. (2022) also consider two encoders $g_1$ and $g_2$, as did previous works on multi-view nonlinear ICA (Gresele et al. 2019; Lyu and Fu, 2020) which we already discuss in our manuscript. However, an **important distinction** is that all of these works assume mutually independent components (Gresele et al. 2019; Lyu and Fu, 2020) or independent groups of shared and view-specific components (Lyu et al. 2022), as we already discuss in Section 6 of the updated manuscript. In contrast, our work rests on a **weaker set of relevant assumptions**, as already described under [major distinction 1](https://openreview.net/forum?id=U_2kuqoTcB&noteId=dBb5rJ_peS5). Specifically, the analysis in Appendix I.1 from Lyu et al. (2022), which uses their Equation (I.2a), **does not imply** that block-identifiability would be satisfied under our more general assumptions.
> > >
> > > In summary, our work uses a similar asymptotic objective but **we prove block-identifiability in a more general setup**, as already discussed under [major distinction 1](https://openreview.net/forum?id=U_2kuqoTcB&noteId=dBb5rJ_peS5). Moreover, our work uses a **different finite-sample estimator** for which we provide **extensive empirical results** supporting block-identifiability in the classical and causal setup, as already discussed in point 3 under [major distinction 2](https://openreview.net/forum?id=U_2kuqoTcB&noteId=dBb5rJ_peS5).
> > >
> > >
> > > > Showing more connections will strengthen this work and reveal more insights.
> > >
> > > We also think that any possible connections will strengthen this work and believe that our previous replies ([1](https://openreview.net/forum?id=U_2kuqoTcB&noteId=dBb5rJ_peS5), [2](https://openreview.net/forum?id=U_2kuqoTcB&noteId=6BWMLsyg-hV), [3](https://openreview.net/forum?id=U_2kuqoTcB&noteId=YEr042dSryI)) make these connections sufficiently clear to all readers.
> > >
> > > We thank the reviewer again for pointing us to the work from Lyu et al. (2022) and for the thoughtful comments. We greatly appreciate the discussion and welcome any further questions.

---

### Author Response · Authors · 2022-11-19
**General Response to All Reviewers**

Dear reviewers,

Thank you all for the constructive feedback.

We greatly appreciate your positive feedback regarding the relevance, quality, and clarity of our work. Most reviewers seem to agree that our work is relevant in that it yields novel insights and important contributions (**YHeS**, **akaD**), which will be helpful for future research on contrastive learning and multimodal learning (**VZnd**, **LEuf**). We acknowledge that reviewer **VsMH** brought to our attention a relevant study that was missing; we already integrated this work into the updated manuscript and further clarify our contribution in the rebuttal. Further, all reviewers highlight the clarity and quality of our work. Reviewers **YHeS** and **VZnd** consider that our theoretical results are backed up by convincing experiments, while reviewers **akaD** and **LEuf** suggest including more realistic experiments and additional ablations---points that we seek to address in the rebuttal.

In summary, in the rebuttal we address the following concerns:
- Reviewer **VsMH**: we acknowledge that we missed a relevant study, which we were not aware of at the time of submission. We clarify our contribution in the rebuttal and have already integrated the study into the updated version of our manuscript.
- Reviewer **akaD**: following your suggestion, we include a new ablation study ([Figure 6](https://ibb.co/WcFFtfK)) where we vary the contribution of style and modality-specific information relative to content. We are pleased to report that the new results corroborate that block-identifiability of content still holds when we vary the relative contribution across these groups of factors; our results even provide some intuition about the sample complexity.
- Reviewer **VZnd**: we clarify one likely misunderstanding that comes up multiple times in your review and explains why our study does not include certain ablations that you have originally suggested. Further, we broaden the scope of our study with an additional experiment on pairs of high-dimensional images ([Figure 10](https://ibb.co/h9rBZnf)).
- Reviewer **LEuf**: to address your concerns, we provide new results that demonstrate how the number of content factors can be estimated in practice ([Figure 7](https://ibb.co/m52thXy)). We also address your question with regard to the convolutional architecture of the text encoder.
- Reviewer **YHeS**: we integrated your suggestions and provide two new experiments to bridge the gap between continuous and discrete data ([Figure 5](https://ibb.co/ZYMGgXF) and [Figure 10](https://ibb.co/h9rBZnf)) as well as a more rigorous evaluation of block-identifiability in the causal setting ([Figure 8](https://ibb.co/M7h0gHD)).

---

### Decision · Program_Chairs · 2023-01-20

**Decision:**

Accept: poster

**Justification For Why Not Higher Score:**

The proof of main theorem largely reuses the proof techniques of von Kugelgen et al. 2021. von Kugelgen et al. 2021 showed the "can not contain style" result without private variables, and allows dependency between content and style. The current paper additionally argued the "can not contain private variables" part, with an argument different from Lyu et al. 2022.

The new development here is, in my opinion, somewhat limited/incremental.



**Justification For Why Not Lower Score:**

Reviewers are generally happy with the results and the presentations.

**Metareview: Summary, Strengths And Weaknesses:**

The paper provides identifiability results for multi-view SSL. The problem setup involves shared variables, style variables, and private variables, and is a combination/generalization of

von Kugelgen et al. 2021: same modality (one generation function), latent space consists of content + style, dependencies between content and style are allowed
Lyu et al. 2022: two modalities (two generation function), private variables are independent from shared ones.
The data generation process combines formulations from both.

The present paper shows the extracted latent representations matched by the two modalities are the shared variables, and thus cannot contain style and cannot contain private variables.

Strength:
The paper studies a setup that generalizes those of previous work, and still achieve identifiability.

Weakness:
While the paper solves additional challenge, the proof techniques mostly follow that of previous work (specifically  von Kugelgen et al. 2021).

**Note From Pc:**

if the above contains the word "oral" or "spotlight" please see: "oral" presentation means -> notable-top-5% and "spotlight" means -> notable-top-25%. As stated in our emails, we are disassociating presentation type from AC recommendations